# Illuminating Protein Function Prediction through Inter-Protein Similarity Modeling

## Abstract

Proteins, central to biological systems, exhibit complex interactions between sequences, structures, and functions shaped by physics and evolution, posing a challenge for accurate function prediction. Recent advances in deep learning techniques demonstrate substantial potential for precise function prediction through learning representations from extensive protein sequences and structures. Nevertheless, practical function annotation heavily relies on modeling protein similarity using sequence or structure retrieval tools, given their accuracy and interpretability. To study the effect of inter-protein similarity modeling, in this paper, we comprehensively benchmark the retriever-based methods against predictors on protein function tasks, demonstrating the potency of retriever-based approaches. Inspired by these findings, we first introduce an innovative variational pseudo-likelihood framework, **ProtIR**, designed to improve function prediction through iterative refinement between predictors and retrievers. ProtIR combines the strengths of both predictors and retrievers, showcasing around 10% improvement over vanilla predictor-based methods. Additionally, it delivers performance on par with protein language model-based methods, yet without the need for massive pre-training, underscoring the efficiency of our framework. We also discover that integrating structural information into protein language model-based retrievers significantly enhances their function annotation capabilities. When ensembled with predictors, this approach achieves top results in two function annotation tasks.

## 1 Introduction

Proteins, being fundamental components in biological systems, hold a central position in a myriad of biological activities, spanning from catalytic reactions to cell signaling processes. The complexity of these macromolecules arises from the intricate interactions between their sequences, structures, and functionalities, influenced by both physical principles and evolutionary processes (Sadowski & Jones, 2009). Despite decades of research, understanding protein function remains a challenge, with a large portion of proteins either lacking characterization or having incomplete understanding of their roles.

Recent progress in Next Generation Sequencing (NGS) technology (Behjati & Tarpey, 2013) and breakthroughs in structure prediction tools (Jumper et al., 2021) have facilitated the accumulation of a vast repository of protein sequences and structures. Harnessing these extensive data, protein representation learning from sequences or structures has emerged as a promising approach for accurate function prediction. Sequence-based methods treat protein sequences as the language of life and train protein language models on billions of natural protein sequences (Elnaggar et al., 2021; 2023; Rives et al., 2021; Lin et al., 2023), while structure-based methods model protein structures as graphs and employ 3D graph neural networks to facilitate message passing between various residues (Gligorijević et al., 2021; Zhang et al., 2023a; Fan et al., 2023).

Despite the impressive performance of machine learning techniques in predicting protein functions, practical function annotation primarily relies on modeling the similarity between different proteins. This is achieved through the use of widely adopted sequence comparison tools such as BLAST (McGinnis & Madden, 2004; Conesa et al., 2005). These tools operate under the evolutionary assumption that proteins with similar sequences likely possess similar functions, offering interpretability by identifying the most closely related reference example for function prediction (Dickson & Mofrad, 2023). Beyond function prediction by retrieving similar sequences,

a probably more plausible assumption is that proteins with similar structures also exhibit similar functions, as protein structures have a more direct influence on determining function (Roy et al., 2015). Recent advancements in structure retrievers (van Kempen et al., 2023), along with progress in structure prediction protocols (Jumper et al., 2021; Lin et al., 2023), have paved the way to explore function prediction methods based on various structure retrievers.

To study the effect of inter-protein similarity modeling, in this paper, we comprehensively benchmark various sequence and structure retriever-based methods against predictor-based approaches on standard protein function annotation tasks, namely Enzyme Commission number and Gene Ontology term prediction. To address the need for robust neural structure retrievers, we introduce a novel strategy wherein we train general protein structure encoders on fold classification tasks, ensuring that the resulting protein representations encapsulate essential structural insights. The experimental results show that retriever-based methods can yield comparable or superior performance compared to predictor-based approaches without massive pre-training. However, it remains a challenge to design a *universal* retriever that can match the state-of-the-art performance of predictor-based methods across *all* functions, regardless of whether the retriever is based on sequences or structures.

Inspired by the principles of retriever-based methods in modeling inter-protein similarity, we introduce two distinct strategies aimed at enhancing function prediction accuracy for predictors, with and without protein language models (PLMs), respectively. We first present an innovative variational pseudo-likelihood framework to model the joint distribution of functional labels across different proteins, ultimately improving predictors without massive pre-training. Utilizing the EM algorithm to optimize the evidence lower bound, we develop an iterative refinement framework that iterates between function predictors and retrievers. This flexible framework, named **ProtIR**, harnesses the advantages of both protein predictors and retrievers and can be applied to any protein encoder. Our experimental results on two state-of-the-art protein structure encoders, GearNet (Zhang et al., 2023a) and CDConv (Fan et al., 2023), clearly demonstrate that the ProtIR framework improves vanilla predictors by an average improvement of approximately 10% across different datasets. Moreover, it achieves comparable performance to protein language model-based methods without large-scale pre-training, underscoring the efficacy of our approach. For enhancing PLM-based methods, we propose a time-efficient alternative. We show that complementing a PLM-based retriever with structural insights makes it better capture protein functional similarity, significantly improving its performance. An ensemble of this enhanced PLM-based retriever and predictor achieves state-of-the-art results in two function annotation tasks, demonstrating the effect of combining inter-protein structural similarity with PLM-based approaches. Our contributions are three-fold:

1. We systematically evaluate retriever- and predictor-based methods and introduce a novel approach for training general protein structure retrievers based on arbitrary protein encoders.
2. We formulate an iterative refinement framework, ProtIR, that operates between predictors and retrievers, significantly enhancing the predictors without massive pre-training.
3. We novelly find that injecting structural details to PLM-based retrievers improves their ability to annotate functions. This method, ensembled with predictors, achieves top results in two tasks.

## 2 FUNCTION PREDICTION WITH RETRIEVER-BASED METHODS

### 2.1 PRELIMINARY

**Proteins.** Proteins are macromolecules formed through the linkage of residues via dehydration reactions and peptide bonds. While only 20 standard residue types exist, their exponential combinations play a pivotal role in the extensive diversity of proteins found in the natural world. The specific ordering of these residues determines the 3D positions of all the atoms within the protein, *i.e.*, the protein structure. Following the common practice, we utilize only the alpha carbon atoms to represent the backbone structure of each protein. Each protein $x$ can be expressed as a pair of a sequence and structure, and is associated with function labels $y \in \{0, 1\}^{n_c}$, where there are $n_c$ distinct functional terms, and each element indicates whether the protein performs a specific function.

**Problem Definition.** In this paper, we delve into the problem of protein function prediction. Given a set of proteins $x_V = x_L \bigcup x_U$ and the labels $y_L$ of a few labeled proteins $L \subset V$, our objective is to predict the labels $y_U$ for the remaining unlabeled set $U = V \backslash L$. Typically, methods based on supervised learning train an encoder denoted as $\psi$ to maximize the log likelihood of the ground truth

labels in the training set, known as predictor-based methods. This optimization can be formulated as:

$$\max_\psi \ \log p_\psi(\boldsymbol{y}_L|\boldsymbol{x}_L) = \sum_{n \in L} \boldsymbol{y}_n \log p_\psi(\boldsymbol{y}_n|\boldsymbol{x}_n) + (\boldsymbol{1} - \boldsymbol{y}_n)\log(\boldsymbol{1} - p_\psi(\boldsymbol{y}_n|\boldsymbol{x}_n)), \quad (1)$$

where $p_\psi(\boldsymbol{y}_n|\boldsymbol{x}_n) = \sigma(\text{MLP}(\psi(\boldsymbol{x}_n)))$, and $\sigma(\cdot)$ represents the sigmoid function. The ultimate goal is to generalize the knowledge learned by the encoder to unlabeled proteins and maximize the likelihood $p_\psi(\boldsymbol{y}_U|\boldsymbol{x}_U)$ for the function labels in the test set.

## 2.2 RETRIEVER-BASED FUNCTION PREDICTION

Despite the success of machine learning in protein function prediction, practical annotation often uses sequence similarity tools like BLAST (Altschul et al., 1997; Conesa et al., 2005). These methods, based on the assumption that similar sequences imply similar functions, offer interpretability by presenting closely related reference examples for function prediction.

These retriever-based methods exhibit a close connection with kernel methods commonly studied in machine learning (Shawe-Taylor & Cristianini, 2003). In this context, the prediction for an unlabeled protein $i \in U$ leverages the labels from the labeled set $L$ through the following expression:

$$\hat{\boldsymbol{y}}_i = \sum_{j \in \mathcal{N}_k(i)} \widetilde{\mathcal{K}}(\boldsymbol{x}_i, \boldsymbol{x}_j) \cdot \boldsymbol{y}_j, \ \text{with} \ \widetilde{\mathcal{K}}(\boldsymbol{x}_i, \boldsymbol{x}_j) = \mathcal{K}(\boldsymbol{x}_i, \boldsymbol{x}_j)/\sum_{t \in \mathcal{N}_k(i)} \mathcal{K}(\boldsymbol{x}_i, \boldsymbol{x}_t) \quad (2)$$

where the kernel function $\mathcal{K}(\cdot, \cdot)$ quantifies the similarity between two proteins, and $\mathcal{N}_k(i) \subset L$ represents the top-$k$ most similar proteins to protein $i$ in the labeled set. For efficiency, we consider only a subset of labeled proteins and re-normalize the similarity within the retrieved set $\mathcal{N}_k(i)$. It is important to note that various methods differ in their specific definitions of the similarity kernel.

## 2.3 NEURAL STRUCTURE RETRIEVER

While sequence retrievers are popular, the assumption that structurally similar proteins share functions is more plausible due to the direct impact of structure on function (Roy et al., 2015). Recent developments in structure retrievers and prediction protocols like AlphaFold2 (Jumper et al., 2021) have opened up promising avenues for exploring various structure-based retrieval methods.

Moving beyond traditional retrievers that compare protein structures in Euclidean space, we adopt advanced protein structure representation learning techniques. Our method uses a protein structure encoder to map proteins into a high-dimensional latent space, where their similarities are measured using cosine similarity. To guarantee that these representations reflect structural information, we pre-train the encoder on a fold classification task (Hou et al., 2018) using 16,712 proteins from 1,195 different folds in the SCOPe 1.75 database (Murzin et al., 1995). This pre-training helps ensure proteins within the same fold are similarly represented.

Formally, our objective is to learn a protein encoder $\phi$ through pre-training on a protein database $\boldsymbol{x}_D$ with associated fold labels $\boldsymbol{c}_D$. The encoder optimization involves maximizing the log likelihood:

$$\max_\phi \ \log p_\phi(\boldsymbol{c}_D|\boldsymbol{x}_D) = \sum_{n \in D} \sum_c [c_n = c] \log p_\phi(c_n = c|\boldsymbol{x}_n). \quad (3)$$

Subsequently, we define the kernel function in (2) as a Gaussian kernel on the cosine similarity:

$$\mathcal{K}(\boldsymbol{x}_i, \boldsymbol{x}_j) = \exp(\cos(\phi(\boldsymbol{x}_i), \phi(\boldsymbol{h}_j))/\tau), \quad (4)$$

where $\tau$ serves as the temperature parameter, controlling the scale of similarity values and is typically set to 0.03 in practice. In this work, we will consider GearNet (Zhang et al., 2023a) and CDConv (Fan et al., 2023) as our choice of encoder $\phi$. A notable advantage of these neural retrievers over traditional methods is their flexibility in fine-tuning for specific functions, as will discuss in next section.

## 3 PROTIR: ITERATIVE REFINEMENT BETWEEN PREDICTOR AND RETRIEVER

Retriever-based methods offer interpretable function prediction, but face challenges in accurately predicting all functions due to the diverse factors influencing protein functions. Predictor-based methods, on the other hand, excel by using labeled data to learn and predict functions for new proteins. To combine the best of both, in this section, we introduce an iterative refinement framework based on the EM algorithm, alternating between function predictors and retrievers. In the E-step, we fix the retriever $\phi$ while allowing the predictor $\psi$ to mimic the labels inferred from the retriever, improving the precision of function prediction with inter-protein similarity. In the M-step, we freeze the predictor $\psi$ and optimize the retriever $\phi$ with the labels inferred from the predictor as the target, effectively

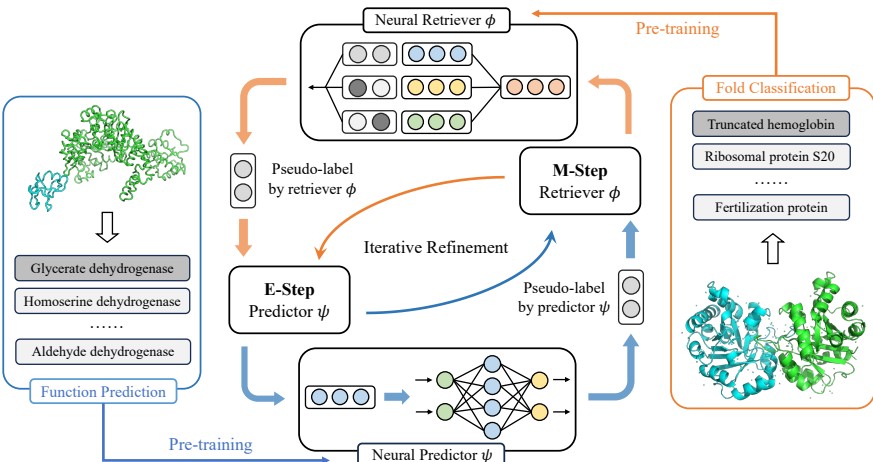

Figure 1: Overview of ProtIR. In the E-step and M-step, the neural predictor $\psi$ and retriever $\phi$ are trained, respectively, and their predictions iteratively refine each other. Before iterative refinement, the predictor $\psi$ and retriever $\phi$ are pre-trained on function prediction and fold classification, respectively.

distilling the predictor's global protein function knowledge into the retriever. This collaborative process mutually strengthens the performance of both the predictor and retriever.

## 3.1 A Pseudolikelihood Variational EM Framework

The effectiveness of retriever-based methods highlights the importance of modeling the relationship between proteins. Therefore, our framework is designed to model the joint distribution of observed function labels given the whole protein set, denoted as $p(\boldsymbol{y}_L|\boldsymbol{x}_V)$. However, directly maximizing this log-likelihood function is challenging due to the presence of unobserved protein function labels. Thus, we opt to optimize the evidence lower bound (ELBO) of the log-likelihood function instead:

$$p(\boldsymbol{y}_L|\boldsymbol{x}_V) \geq \mathbb{E}_{q(\boldsymbol{y}_U|\boldsymbol{x}_U)}[\log p(\boldsymbol{y}_L, \boldsymbol{y}_U|\boldsymbol{x}_V) - \log q(\boldsymbol{y}_U|\boldsymbol{x}_U)], \tag{5}$$

where $q(\boldsymbol{y}_U|\boldsymbol{x}_U)$ denotes a proposal distribution over $\boldsymbol{y}_U$. The equality is achieved when the proposal distribution aligns with the posterior distribution, i.e., $q(\boldsymbol{y}_U|\boldsymbol{x}_U) = p(\boldsymbol{y}_U|\boldsymbol{y}_L, \boldsymbol{x}_V)$.

The ELBO is maximized through alternating optimization between the model distribution $p$ (M-step) and the proposal distribution $q$ (E-step). In the M-step, we keep the distribution $q$ fixed and optimize the retriever-based distribution $p$ to maximize the log-likelihood function. However, direct optimization involves calculating the partition function in $p$, which is computationally intensive. To circumvent this, we optimize the pseudo-likelihood function (Besag, 1975):

$$\mathbb{E}_{q(\boldsymbol{y}_U|\boldsymbol{x}_U)}[\log p(\boldsymbol{y}_L, \boldsymbol{y}_U|\boldsymbol{x}_V)] \approx \mathbb{E}_{q(\boldsymbol{y}_U|\boldsymbol{x}_U)}[\sum_{n \in V} \log p(\boldsymbol{y}_n|\boldsymbol{x}_V, \boldsymbol{y}_{V \setminus n})] \tag{6}$$

In the E-step, we hold the distribution $p$ fixed and optimize $q$ to minimize the KL divergence $\text{KL}(q(\boldsymbol{y}_U|\boldsymbol{x}_U)||p(\boldsymbol{y}_U|\boldsymbol{x}_V, \boldsymbol{y}_L))$, aiming to tighten the lower bound.

## 3.2 Parameterization

We now discuss how to parameterize the distributions $p$ and $q$ with retrievers and predictors, respectively. For the proposal distribution $q$, we adopt a mean-field assumption, assuming independence among function labels for different proteins. This leads to the factorization:

$$q_\psi(\boldsymbol{y}_U|\boldsymbol{x}_U) = \prod_{n \in U} q_\psi(\boldsymbol{y}_n|\boldsymbol{x}_n), \tag{7}$$

where each term $q_\psi(\boldsymbol{y}_n|\boldsymbol{x}_n)$ is parameterized using an MLP head applied to the representations outputted from a protein encoder $\psi$ as introduced in Sec. 2.1.

On the other hand, the conditional distribution $p(\boldsymbol{y}_n|\boldsymbol{x}_V, \boldsymbol{y}_{V \setminus n})$ aims to utilize the protein set $\boldsymbol{x}_V$ and other node labels $\boldsymbol{y}_{V \setminus n}$ to characterize the label distribution of each protein $n$. This formulation aligns naturally with a retriever-based method by retrieving similar proteins from the labeled set. Hence, we model $p_\phi(\boldsymbol{y}_n|\boldsymbol{x}_V, \boldsymbol{y}_{V \setminus n})$ with a retriever $\phi$ as in (2) and (4) to effectively model the

relationship between different proteins. In the following sections, we elaborate on the optimization of both the predictor distribution $q_\psi$ and the retriever distribution $p_\phi$.

## 3.3 E-STEP: PREDICTOR OPTIMIZATION

In the E-step, we keep the retriever $\phi$ fixed and optimize the predictor $\psi$ to maximize the evidence lower bound, allowing the retriever's understanding of global protein relationships to be distilled into the predictor. The goal is to minimize the KL divergence between the proposal distribution and the posterior distribution, expressed as , $\text{KL}(q_\psi(\boldsymbol{y}_U|\boldsymbol{x}_U)||p_\phi(\boldsymbol{y}_U|\boldsymbol{x}_V, \boldsymbol{y}_L))$. Directly optimizing this divergence proves challenging due to the reliance on the entropy of $q_\psi(\boldsymbol{y}_U|\boldsymbol{x}_U)$, the gradient of which is difficult to handle. To circumvent this, we adopt the wake-sleep algorithm (Hinton et al., 1995) to minimize the reverse KL divergence, leading to the following objective function to maximize:

$$-\text{KL}(p_\phi(\boldsymbol{y}_U|\boldsymbol{x}_V, \boldsymbol{y}_L)||q_\psi(\boldsymbol{y}_U|\boldsymbol{x}_U)) = \mathbb{E}_{p_\phi(\boldsymbol{y}_U|\boldsymbol{x}_V, \boldsymbol{y}_L)}[\log q_\psi(\boldsymbol{y}_U|\boldsymbol{x}_U)] + \text{const} \tag{8}$$

$$= \sum_{n \in U} \mathbb{E}_{p_\phi(\boldsymbol{y}_n|\boldsymbol{x}_V, \boldsymbol{y}_L)}[\log q_\psi(\boldsymbol{y}_n|\boldsymbol{x}_n)] + \text{const}, \tag{9}$$

where const denotes the terms irrelevant with $\psi$. This is more tractable as it avoids the need for the entropy of $q_\psi(\boldsymbol{y}_U|\boldsymbol{x}_U)$. To sample from the distribution $p_\phi(\boldsymbol{y}_U|\boldsymbol{x}_V, \boldsymbol{y}_L)$, we annotate the unlabeled proteins by employing $\phi$ to retrieve the most similar proteins from the labeled set using (2). Additionally, the labeled proteins can be used to train the predictor and prevent catastrophic forgetting (McCloskey & Cohen, 1989). Combining this with the pseudo-labeling objective, we arrive at the final objective function for training the predictor:

$$\boxed{E\text{-step:} \quad \max_\psi \ \sum_{n \in U} \mathbb{E}_{p_\phi(\boldsymbol{y}_n|\boldsymbol{x}_V, \boldsymbol{y}_L)}[\log q_\psi(\boldsymbol{y}_n|\boldsymbol{x}_n)] + \sum_{n \in L} \log q_\psi(\boldsymbol{y}_n|\boldsymbol{x}_n).} \tag{10}$$

Intuitively, the second term is a supervised training objective, and the first term acts as a knowledge distillation process, making the predictor align with the label distribution from the retriever.

## 3.4 M-STEP: RETRIEVER OPTIMIZATION

In the M-step, our objective is to keep the predictor $\psi$ fixed and fine-tune the retriever $\phi$ to maximize the pseudo-likelihood, as introduced in (6). Similar to Sec. 3.3, we sample the pseudo-labels $\hat{\boldsymbol{y}}_U$ from the predictor distribution $q_\psi$ for unlabeled proteins. Consequently, the pseudo-likelihood objective can be reformulated as follows:

$$\sum_{n \in U} \log p_\phi(\hat{\boldsymbol{y}}_n|\boldsymbol{x}_V, \boldsymbol{y}_L, \hat{\boldsymbol{y}}_{U \setminus n}) + \sum_{n \in L} \log p_\phi(\boldsymbol{y}_n|\boldsymbol{x}_V, \boldsymbol{y}_{L \setminus n}, \hat{\boldsymbol{y}}_U). \tag{11}$$

Again, the first term represents a knowledge distillation process from the predictor to the retriever via all the pseudo-labels, while the second is a straightforward supervised loss involving observed labels.

The optimization of the retriever distribution $p_\phi$ involves learning the kernel functions $\mathcal{K}(\cdot, \cdot)$ by aligning representations of proteins with identical function labels and pushing apart those with different labels. One potential approach to the problem is supervised contrastive learning (Khosla et al., 2020). However, defining and balancing positive and negative samples in contrastive learning becomes challenging when dealing with the multiple binary labels in (11). To simplify the training of the retriever $\phi$, we transform the contrastive learning into a straightforward multiple binary classification problem akin to the predictor $\psi$. We accomplish this by introducing an MLP head over the representations outputted by $\phi$, denoted as $\tilde{p}_\phi(\boldsymbol{y}_n|\boldsymbol{x}_n) = \sigma(\text{MLP}(\phi(\boldsymbol{x}_n)))$ and optimize it using binary cross entropy loss as outlined in (1). Formally, the M-step can be expressed as:

$$\boxed{M\text{-step:} \quad \max_\phi \ \sum_{n \in U} \log \tilde{p}_\phi(\hat{\boldsymbol{y}}_n|\boldsymbol{x}_n) + \sum_{n \in L} \log \tilde{p}_\phi(\boldsymbol{y}_n|\boldsymbol{x}_n).} \tag{12}$$

By training the model for binary classification, proteins with similar function labels are assigned with similar representations, enhancing the distinction between various function classes. During inference, we integrate the trained retriever $\phi$ back into the orignial formulation in (2).

Finally, the workflow of the EM algorithm is summarized in Fig. 1 and Alg. 1. In practice, we start from a pre-trained predictor $q_\psi$ using labeled function data as in (1) and a retriever $p_\phi$ infused with structrual information from the fold classfication task as in (3). We use validation performance as a criterion for tuning hyperparameters and early stopping. The iterative refinement process typically converges within five rounds, resulting in minimal additional training time.

## 4  RELATED WORK

**Protein Representation Learning.** Previous research focuses on learning protein representations from diverse modalities, including sequences (Lin et al., 2023), multiple sequence alignments (Rao et al., 2021), and structures (Zhang et al., 2023a). Sequence-based methods treat protein sequences as the fundamental language of life, pre-training large models on billions of sequences (Rao et al., 2019; Elnaggar et al., 2021; Rives et al., 2021). Structure-based methods capture different levels of protein structures, including residue-level (Gligorijević et al., 2021; Zhang et al., 2023a), atom-level structures (Jing et al., 2021; Hermosilla et al., 2021), and protein surfaces (Gainza et al., 2020). Diverse self-supervised learning algorithms have been developed to pre-train structure encoders, such as contrastive learning (Zhang et al., 2023a), self-prediction (Chen et al., 2022), denoising score matching (Guo et al., 2022), and diffusion (Zhang et al., 2023c). Recent efforts have been devoted to integrating sequence- and structure-based methods (Wang et al., 2022; Zhang et al., 2023b).

**Retriever-Based Methods.** Retriever-based methods, starting with the k-nearest neighbors (k-NN) approach (Fix & Hodges, 1989; Cover & Hart, 1967), represent a critical paradigm in the field of machine learning and information retrieval, with application in text (Khandelwal et al., 2020; Borgeaud et al., 2021), image (Papernot & Mcdaniel, 2018; Borgeaud et al., 2021), and video generation (Jin et al., 2023). Designing protein retrievers to capture similar evolutionary and structural information has been an important topic for decades (Chen et al., 2018). These retrievers can be employed for improving function annotation (Conesa et al., 2005; Ma et al., 2023; Yu et al., 2023).

In this study, we take the first systematic evaluation of modern methods from both categories for function annotation. Different from existing works, we develop a simple strategy to train a general neural structure retriever. Moreover, we propose a novel iterative refinement framework to combine the predictor- and retriever-based methods, maximizing the utility of scarce function labels.

## 5  EXPERIMENTS

In this section, we address two main research questions: the advantages of both predictor- and retriever-based methods, and how retriever-based insights can enhance predictor-based methods. To tackle these questions, experiments are conducted on function annotation tasks (see Sec. 5.1). For the first question, we benchmark standard baselines from both approaches (Sec. 5.2). For the second, we explore incorporating inter-protein similarity in predictors, first by applying the ProtIR framework to pre-trained predictors without pre-training (Sec. 5.3), and then by adding structural information to predictors with protein language models (Sec. 5.4).

### 5.1  EXPERIMENTAL SETUP

We evaluate the methods using two function annotation tasks in Gligorijević et al. (2021). The first task, **Enzyme Commission (EC) prediction**, involves predicting the EC numbers for proteins, indicating their role in biochemical reactions, focusing on the third and fourth levels of the EC tree (Webb et al., 1992). The second task, **Gene Ontology (GO) prediction**, determines if a protein is associated with specific GO terms, classifying them into molecular function (MF), biological process (BP), and cellular component (CC) categories, each reflecting different aspects of protein function.

To ensure a rigorous evaluation, we follow the multi-cutoff split methods outlined in Gligorijević et al. (2021). Specifically, we ensure that the test set only contain PDB chains with a sequence identity of no more than 30%, 50%, and 95% to the training set, aligning with the approach used in Wang et al. (2022). The evaluation of performance is based on the protein-centric maximum F-score, denoted as $F_{max}$, a commonly used metric in the CAFA challenges (Radivojac et al., 2013). Details in App. C.

### 5.2  BENCHMARK RESULTS OF PREDICTOR- AND RETRIEVER-BASED METHODS

**Baselines.** We select two categories of predictor-based baselines for comparison: (1) *Protein Encoders without Pre-training*: This category includes four sequence-based encoders (CNN, ResNet, LSTM and Transformer (Rao et al., 2019)) and three structure-based encoders (GCN (Kipf & Welling, 2017), GearNet (Zhang et al., 2023a), CDConv (Fan et al., 2023)). (2) *Protein Encoders with Massive Pre-training*: This includes methods based on protein language models (PLM) pre-trained on millions to billions of protein sequences, such as DeepFRI (Gligorijević et al., 2021), ProtBERT-BFD (Elnaggar et al., 2021), ESM-2 (Lin et al., 2023) and PromptProtein (Wang et al., 2023). Due to computational constraints, we exclude ESM-2-3B and ESM-2-15B from the benchmark.

Table 1: $F_{max}$ on EC and GO prediction with predictor- and retriever-based methods.

| Method | PLM | EC 30% | EC 50% | EC 95% | GO-BP 30% | GO-BP 50% | GO-BP 95% | GO-MF 30% | GO-MF 50% | GO-MF 95% | GO-CC 30% | GO-CC 50% | GO-CC 95% |
|---|---|---|---|---|---|---|---|---|---|---|---|---|---|
| **Predictor-Based** | | | | | | | | | | | | | |
| CNN | ✗ | 0.366 | 0.372 | 0.545 | 0.197 | 0.197 | 0.244 | 0.238 | 0.256 | 0.354 | 0.258 | 0.260 | 0.387 |
| ResNet | | 0.409 | 0.450 | 0.605 | 0.230 | 0.234 | 0.280 | 0.282 | 0.308 | 0.405 | 0.277 | 0.280 | 0.304 |
| LSTM | | 0.247 | 0.270 | 0.425 | 0.194 | 0.195 | 0.225 | 0.223 | 0.245 | 0.321 | 0.263 | 0.269 | 0.283 |
| Transformer | | 0.167 | 0.175 | 0.238 | 0.267 | 0.262 | 0.264 | 0.184 | 0.195 | 0.211 | 0.378 | 0.388 | 0.405 |
| GCN | | 0.245 | 0.246 | 0.320 | 0.251 | 0.248 | 0.252 | 0.180 | 0.187 | 0.195 | 0.318 | 0.320 | 0.329 |
| GearNet | | **0.700** | **0.769** | **0.854** | 0.348 | 0.359 | 0.406 | 0.482 | 0.525 | 0.613 | 0.407 | 0.418 | 0.458 |
| CDConv | | 0.634 | 0.702 | 0.820 | **0.381** | **0.401** | **0.453** | **0.533** | **0.577** | **0.654** | **0.428** | **0.440** | **0.479** |
| DeepFRI | ✓ | 0.470 | 0.545 | 0.631 | 0.361 | 0.371 | 0.399 | 0.374 | 0.409 | 0.465 | 0.440 | 0.444 | 0.460 |
| ProtBERT-BFD | | 0.691 | 0.752 | 0.838 | 0.308 | 0.321 | 0.361 | 0.497 | 0.541 | 0.613 | 0.287 | 0.293 | 0.308 |
| ESM-2-650M | | 0.763 | 0.816 | 0.877 | 0.423 | 0.438 | 0.484 | 0.563 | 0.604 | 0.661 | 0.497 | 0.509 | 0.535 |
| PromptProtein | | 0.765 | 0.823 | 0.888 | 0.439 | 0.453 | 0.495 | 0.577 | 0.600 | 0.677 | 0.532 | 0.533 | 0.551 |
| **Retriever-Based** | | | | | | | | | | | | | |
| MMseqs | ✗ | 0.781 | 0.833 | 0.887 | 0.323 | 0.359 | 0.444 | 0.502 | 0.557 | 0.647 | 0.237 | 0.255 | 0.332 |
| BLAST | | 0.740 | 0.806 | 0.872 | 0.344 | 0.373 | 0.448 | 0.505 | 0.557 | 0.640 | 0.275 | 0.284 | 0.347 |
| PSI-BLAST | | 0.642 | 0.705 | 0.798 | 0.341 | 0.364 | 0.442 | 0.433 | 0.482 | 0.573 | 0.354 | 0.365 | 0.420 |
| TMAlign | | 0.674 | 0.744 | 0.817 | 0.403 | 0.426 | 0.480 | 0.487 | 0.533 | 0.597 | 0.410 | 0.424 | 0.456 |
| Foldseek | | 0.781 | 0.834 | 0.885 | 0.328 | 0.359 | 0.435 | 0.525 | 0.573 | 0.651 | 0.245 | 0.254 | 0.312 |
| Progres | | 0.535 | 0.634 | 0.727 | 0.353 | 0.379 | 0.448 | 0.428 | 0.480 | 0.573 | 0.374 | 0.390 | 0.438 |
| GearNet *w/ struct.* | | 0.671 | 0.744 | 0.822 | 0.391 | 0.419 | 0.482 | 0.497 | 0.548 | 0.626 | 0.377 | 0.387 | 0.434 |
| CDConv *w/ struct.* | | 0.719 | 0.784 | 0.843 | 0.409 | 0.434 | 0.494 | 0.536 | 0.584 | 0.661 | 0.387 | 0.397 | 0.438 |
| ESM-2-650M | ✓ | 0.585 | 0.656 | 0.753 | **0.398** | **0.415** | **0.477** | 0.462 | 0.510 | 0.607 | **0.427** | **0.436** | **0.472** |
| TM-Vec | | **0.676** | **0.745** | **0.817** | 0.377 | 0.399 | 0.461 | 0.552 | 0.593 | 0.663 | 0.328 | 0.328 | 0.369 |

[*] **Red**: the best results among all; **blue**: the second best results among all; **bold**: the best results within blocks.
[†] Two proposed neural retrievers are denoted as GearNet *w/ struct.* and CDConv *w/ struct.*, respectively.

For retriever-based methods, we considered retrievers with and without protein language models. For those without PLMs, we select three sequence retrievers, MMSeqs (Steinegger & Söding, 2017), BLAST (Altschul et al., 1990) and PSI-BLAST (Altschul et al., 1997), and three structure retrievers, TMAlign (Zhang & Skolnick, 2005), Foldseek (van Kempen et al., 2023) and Progres (Greener & Jamali, 2022). Additionally, we train two neural structure retrievers by using GearNet and CDConv on fold classification tasks as in (3). For retrievers with PLMs, we consider using ESM-2-650M (Lin et al., 2023) and recently proposed TM-Vec (Hamamsy et al., 2023) for retrieving similar proteins.

**Training.** For predictor-based methods, except for GearNet and ESM-2-650M, all results were obtained from a previous benchmark (Zhang et al., 2023a). We re-implement GearNet, optimizing it following CDConv's implementation with a 500-epoch training, leading to significant improvements over the original paper (Zhang et al., 2023a). For ESM-2-650M, we fine-tune the model for 50 epochs. For GearNet and CDConv retrievers, we train them on the fold dataset for 500 epochs, selecting the checkpoint with the best validation performance as the final retrievers. Detailed training setup for other retriever-based methods is provided in App. E.2. All these models are trained on 1 A100 GPU.

**Results.** The benchmark results are presented in Table 1. Here is an analysis of the findings[1]:

*Firstly, retriever-based methods exhibit comparable or superior performance to predictor-based methods without pre-training.* A comparison between the first and third blocks in Table 1 reveals that retrievers can outperform predictors even without training on function labels. This supports the hypothesis that proteins sharing evolutionary or structural information have similar functions.

*Secondly, when fine-tuned, predictor-based methods using Protein Language Models (PLMs) significantly outperform retrievers.* This aligns with the principle that deep learning techniques efficiently leverage large pre-training datasets, enabling neural predictors to capture more evolutionary information than traditional, hard-coded retrievers.

*Thirdly, contrary to expectations, structure retrievers do not always outperform sequence retrievers.* As shown in the third block of the table, sequence retrievers like MMSeqs and BLAST perform better than structure retrievers like GearNet and CDConv on EC tasks but are less effective for GO tasks. This discrepancy may underscore the importance of evolutionary information for enzyme catalysis, while structural aspects are more crucial for molecular functions.

*Fourthly, a universal retriever excelling across all functions is still lacking.* For instance, the best structure retriever, CDConv, underperforms in EC number predictions, whereas sequence retrievers struggle with GO predictions. This suggests that different functions rely on varying factors, which may not be fully captured by these general-purpose sequence and structure retrievers.

---

[1]Notably, the results for GO-CC differ significantly from other tasks. GO-CC aims to predict the cellular compartment where the protein functions, which is less directly related to the protein's function itself.

Table 2: $F_{max}$ on EC and GO prediction with iterative refinement and transductive learning baselines.

| Model | Method | EC | | | GO-BP | | | GO-MF | | | GO-CC | | |
|---|---|---|---|---|---|---|---|---|---|---|---|---|---|
| | | 30% | 50% | 95% | 30% | 50% | 95% | 30% | 50% | 95% | 30% | 50% | 95% |
| GearNet | Predictor | 0.700 | 0.769 | 0.854 | 0.348 | 0.359 | 0.406 | 0.482 | 0.525 | 0.613 | 0.407 | 0.418 | 0.458 |
| | Pseudo-labeling | 0.699 | 0.767 | 0.852 | 0.344 | 0.355 | 0.403 | 0.490 | 0.532 | 0.617 | 0.420 | 0.427 | 0.466 |
| | Temporal ensemble | 0.698 | 0.765 | 0.850 | 0.339 | 0.348 | 0.399 | 0.480 | 0.526 | 0.613 | 0.402 | 0.412 | 0.454 |
| | Graph conv network | 0.658 | 0.732 | 0.817 | 0.379 | 0.395 | 0.443 | 0.479 | 0.528 | 0.609 | 0.437 | 0.452 | 0.483 |
| | **ProtIR** | **0.743** | **0.810** | **0.881** | **0.409** | **0.431** | **0.488** | **0.518** | **0.564** | **0.650** | **0.439** | **0.452** | **0.501** |
| | **Improvement ↑** | **6.1%** | **5.3%** | **3.1%** | **17.5%** | **20.0%** | **20.1%** | **7.4%** | **7.4%** | **6.0%** | **7.8%** | **8.1%** | **9.3%** |
| CDConv | Predictor | 0.634 | 0.702 | 0.820 | 0.381 | 0.401 | 0.453 | 0.533 | 0.577 | 0.654 | 0.428 | 0.440 | 0.479 |
| | Pseudo-labeling | 0.722 | 0.784 | 0.861 | 0.397 | 0.413 | 0.465 | 0.529 | 0.573 | 0.653 | 0.445 | 0.458 | 0.495 |
| | Temporal ensemble | 0.721 | 0.785 | 0.862 | 0.381 | 0.394 | 0.446 | 0.523 | 0.567 | 0.647 | 0.444 | 0.455 | 0.492 |
| | Graph conv network | 0.673 | 0.742 | 0.818 | 0.380 | 0.399 | 0.455 | 0.496 | 0.545 | 0.627 | 0.417 | 0.429 | 0.465 |
| | **ProtIR** | **0.769** | **0.820** | **0.885** | **0.434** | **0.453** | **0.503** | **0.567** | **0.608** | **0.678** | **0.447** | **0.460** | **0.499** |
| | **Improvement ↑** | **21.2%** | **16.8%** | **4.2%** | **13.9%** | **12.9%** | **23.8%** | **6.3%** | **5.3%** | **3.6%** | **4.4%** | **4.5%** | **4.1%** |
| PromptProtein | | 0.765 | 0.823 | 0.888 | 0.439 | 0.453 | 0.495 | 0.577 | 0.600 | 0.677 | 0.532 | 0.533 | 0.551 |

\* **Red**: >20% improvement; **blue**: 10%-20% improvement; **bold**: 3%-10% improvement.

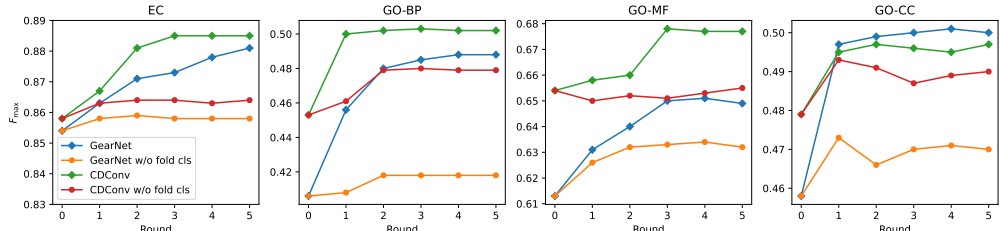

Figure 2: $F_{max}$ on function annotation tasks *vs.* number of rounds in iterative refinement. Besides the default iterative refinement models, we also depict curves for models with a retriever not pre-trained on fold classification, highlighting the impact of incorporating structural information.

In conclusion, while retriever-based methods demonstrate potential for accurate function prediction without extensive pre-training, a universal retriever with state-of-the-art performance across all functions is yet to be developed. Nonetheless, the concept of inter-protein similarity modeling shows potential for enhancing function annotation accuracy, as will be shown in next section.

## 5.3 RESULTS OF ITERATIVE REFINEMENT FRAMEWORK

**Setup.** We employ the GearNet and CDConv trained on EC and GO as our backbone model and conduct a comprehensive evaluation by comparing our proposed iterative refinement framework with several baseline methods. As the iterative refinement framework falls under the category of transductive learning (Vapnik, 2000), we benchmark our approach against three well-established deep semi-supervised learning baselines: pseudo-labeling (Lee, 2013), temporal ensemble (Laine & Aila, 2017), and graph convolutional networks (Kipf & Welling, 2017). These baselines are trained for 50 epochs. For our method, we iterate the refinement process for up to 5 rounds, halting when no further improvements are observed. In each iteration, both the E-step and M-step are trained for 30 epochs.

**Results.** The results are summarized in Table 2. Notably, our proposed iterative refinement consistently demonstrates substantial improvements across various tasks and different backbone models when compared to both vanilla predictors and other transductive learning baselines. On average, GearNet showcases a remarkable improvement of 9.84%, while CDConv exhibits an impressive 10.08% enhancement, underscoring the effectiveness of our approach.

Moreover, in comparison to the state-of-the-art predictor-based method, PromptProtein, CDConv achieves similar performance on EC, GO-BP, and GO-MF tasks while reducing the need for pre-training on millions of sequences. Our method demands less than 24 hours on a single GPU for additional training (500 epochs for retriever and 300 epochs for refinement, taking 1 minute per epoch), whereas pre-training a protein language model typically costs thousands of GPU hours. This efficiency underscores the practicality of our approach for maximizing the utility of limited data.

**Analysis and Ablation Study.** To analyze the iterative refinement process, we present the test performance curve as a function of the number of rounds in Fig. 2. The results reveal a consistent enhancement in test performance for both models, with convergence typically occurring within five

Table 3: $F_{max}$ on EC and GO prediction with predictors and retrievers based on PLMs.

| | Method | EC | | | GO-BP | | | GO-MF | | | GO-CC | | |
|---|---|---|---|---|---|---|---|---|---|---|---|---|---|
| | | 30% | 50% | 95% | 30% | 50% | 95% | 30% | 50% | 95% | 30% | 50% | 95% |
| **Predictor** | ESM-2-8M | 0.510 | 0.565 | 0.658 | 0.323 | 0.331 | 0.368 | 0.395 | 0.427 | 0.502 | 0.417 | 0.431 | 0.457 |
| | ESM-2-35M | 0.678 | 0.744 | 0.818 | 0.382 | 0.393 | 0.443 | 0.493 | 0.533 | 0.610 | 0.444 | 0.457 | 0.481 |
| | ESM-2-150M | 0.749 | 0.802 | 0.865 | 0.397 | 0.413 | 0.460 | 0.558 | 0.599 | 0.667 | 0.481 | 0.493 | 0.523 |
| | ESM-2-650M | **0.763** | **0.816** | **0.877** | **0.423** | **0.438** | **0.484** | **0.563** | **0.604** | **0.661** | **0.497** | **0.509** | **0.535** |
| **Retriever** | ESM-2-8M | 0.423 | 0.449 | 0.581 | 0.337 | 0.355 | 0.423 | 0.420 | 0.455 | 0.553 | 0.359 | 0.367 | 0.413 |
| | ESM-2-35M | 0.428 | 0.436 | 0.560 | 0.390 | 0.411 | 0.471 | **0.485** | **0.531** | **0.618** | 0.402 | 0.410 | 0.448 |
| | ESM-2-150M | 0.482 | 0.538 | 0.656 | 0.383 | 0.404 | 0.468 | 0.467 | 0.516 | 0.611 | 0.415 | 0.427 | 0.462 |
| | ESM-2-650M | **0.585** | **0.656** | **0.753** | **0.398** | **0.415** | **0.477** | 0.462 | 0.510 | 0.607 | **0.427** | **0.436** | **0.472** |
| | **ESM-2-8M** *w/ struct.* | 0.482 | 0.499 | 0.620 | 0.368 | 0.389 | 0.453 | 0.461 | 0.504 | 0.596 | 0.377 | 0.392 | 0.433 |
| | **ESM-2-35M** *w/ struct.* | 0.502 | 0.553 | 0.658 | 0.417 | 0.439 | 0.494 | 0.522 | 0.570 | 0.649 | 0.414 | 0.425 | 0.459 |
| | **ESM-2-150M** *w/ struct.* | 0.547 | 0.598 | 0.690 | 0.434 | 0.455 | 0.506 | 0.548 | 0.594 | 0.666 | 0.424 | 0.436 | 0.472 |
| | **ESM-2-650M** *w/ struct.* | **0.676** | **0.742** | **0.817** | **0.455** | **0.472** | **0.519** | **0.570** | **0.612** | **0.678** | **0.448** | **0.455** | **0.485** |
| | PromptProtein | 0.765 | 0.823 | 0.888 | 0.439 | 0.453 | 0.495 | 0.577 | 0.600 | 0.677 | 0.532 | 0.533 | 0.551 |
| | **ESM-2-650M ensemble** | 0.768 | 0.819 | 0.879 | 0.459 | 0.472 | 0.516 | 0.588 | 0.627 | 0.690 | 0.506 | 0.514 | 0.540 |

\* **Red**: the best results among all; **blue**: the second best results among all; **bold**: the best results within blocks.

rounds. This underscores the efficiency of our iterative framework in yielding performance gains relatively swiftly. Additionally, we examine the impact of injecting structural information into the retriever by comparing results with and without a fold classification pre-trained retriever. Notably, while improvements are observed without fold pre-training, the performance is significantly superior with this pre-training, emphasizing the importance of incorporating structural insights.

## 5.4 Injecting Structural Information Into Protein Language Models

While effective, our iterative refinement relies on structure encoders and requires structures as input, posing a challenge for datasets lacking such structural information. Furthermore, the process of fine-tuning Protein Language Model (PLM)-based predictors through multiple iterations, as outlined in ProtIR, can be notably time-consuming. To address these limitations, we investigate an alternative approach to enhance PLM-based predictors. We first pre-train a PLM-based retriever that incorporates structural insights by pre-training the models on fold classification, as suggested in (3). Then, this retriever is ensemble with the corresponding PLM-based predictor by taking the average of their prediction. We employ various sizes of ESM-2 as backbone models and assess their performance when used as predictors and retrievers for function prediction. The results are presented in Table 3, with ESM-2 models incorporating fold classification pre-training denoted as ESM-2 *w/ struct.*

In Table 3, a comparison between the second and third blocks highlights a significant boost in performance for all examined PLM-based retrievers by incorporating structural information. This presents a potential solution for enhancing protein language models. Notably, this method outperforms predictor-based methods in GO-BP and GO-MF tasks, albeit showing slightly lower performance in EC and GO-CC. This shows the distinct nature of protein functions and suggests that the efficacy of retriever-based methods should not overshadow the essential role of predictor-based approaches. After ensembling the ESM-2-650M-based predictor and retriever, we are able to further improve the predictor's performance easily and achieve the state-of-the-art performance on GO-BP and GO-MF.

## 6 Conclusion

In this study, we comprehensively evaluated various sequence and structure retriever-based methods against predictor-based approaches for protein function annotation tasks. We introduced a novel training strategy by training general protein structure encoders on fold classification tasks, to build neural structure retrievers. Our experimental results revealed that retriever-based methods, even without extensive pre-training, could rival or surpass predictor-based approaches using protein language models. We further introduced a novel framework, named ProtIR, significantly enhancing function prediction accuracy by modeling inter-protein similarity. The ProtIR framework, harnessing predictor and retriever advantages, demonstrated substantial performance improvements and efficiency compared to state-of-the-art methods. Our discovery also reveals that complementing protein language models retrievers with structural insights can greatly boost the accuracy. Future works include the application on other protein tasks, *e.g.*, protein engineering and docking.

## REPRODUCIBILITY STATEMENT

For reproducibility, we provide the implementation details for all baselines and our methods in Section 5 and Appendix D. Specifically, for benchmarking retriever-based methods, the configuration of retrievers and prediction methods can be found in Sections 5.1 and 5.2. The pseudo-code of ProtIR are provided in Appendix B and the training details of are given in 5.3. The source code of the paper will be released upon acceptance.

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

## A    MORE RELATED WORK AND BROADER IMPACT

**Protein Retriever.** In the domain of proteins, retriever-based methods have long been employed for function annotation (Conesa et al., 2005), utilizing both sequence (Altschul et al., 1990; Melvin et al., 2011; Buchfink et al., 2021; Hamamsy et al., 2023) and structure-based approaches (Shindyalov & Bourne, 1998; Yang & Tung, 2006; Zhao et al., 2013; Holm, 2019; Trinquier et al., 2022; Greener & Jamali, 2022; van Kempen et al., 2023). Recent endeavors have extended retrievers to retrieve similar sequences from expansive databases, augmenting inputs and subsequently enhancing function prediction performance (Ma et al., 2023; Zou et al., 2023; Dickson & Mofrad, 2023; Kilinc et al., 2023; Chen et al., 2023). Instead of designing a new protein retriever, our work proposes a general strategy to train a neural structure retriever and studies how to use the idea of inter-protein similarity modeling to improve function annotation accruacy.

**Protein Network Propagation for Function Prediction.** Besides directly measuring inter-protein similarities based on sequences and structures, there is a parallel line of research that focuses on function annotation through protein-protein interaction (PPI) networks, exemplified by tools like STRING (Szklarczyk et al., 2019). These networks map both direct physical and indirect functional interactions among proteins. Recent approaches in this domain involve functional label propagation within these networks (Mostafavi et al., 2008; Wang et al., 2017; You et al., 2019; Cho et al., 2016; Kulmanov et al., 2018; Yao et al., 2021), and adapting these methods to PPI networks of newly sequenced species (You et al., 2021; Torres et al., 2021). However, a key limitation of these methods is that they are not able to make predictions for newly sequenced proteins absent in existing PPI networks. Moreover, knowing protein-protein interactions is essentially a more difficult challenge, as it requires a more comprehensive understanding of protein properties. These problems make this line of work hard to use in real-world settings.

**Transductive Learning.** Our iterative refinement framework falls into the category of transductive learning, focusing on optimizing performance for specific sets of interest rather than reasoning general rules applicable to any test cases (Vapnik, 2006). Typical transductive learning methods encompass generative techniques (Springenberg, 2015; Kingma et al., 2014), consistency regularization approaches (Rasmus et al., 2015; Laine & Aila, 2017), graph-based algorithms (Kipf & Welling, 2017; Gilmer et al., 2017), pseudo-labeling strategies (Lee, 2013), and hybrid methodologies (Verma et al., 2022; Berthelot et al., 2019). In contrast to existing approaches, our work develops a novel iterative refinement framework for mutual enhancement between predictors and retrievers.

**Broader Impact and Ethical Considerations.** The main objective of this research project is to enable more accurate protein function annotation by modeling inter-protein similarity. Unlike traditional protein retrievers, our approach utilizes structural information in the CATH dataset to build a neural structure retriever. This advantage allows for more comprehensive analysis of protein research and holds potential benefits for various real-world applications, including protein optimization, sequence and structure design. It is important to acknowledge that powerful function annotation models can potentially be misused for harmful purposes, such as the design of dangerous drugs. We anticipate that future studies will address and mitigate these concerns.

**Limitations.** In this study, we explore the design of a general neural structure retriever and conduct benchmarks on existing retrievers and predictors for function annotation. However, given the extensive history of protein retriever development in the bioinformatics field, it is impractical to include every retriever in our benchmark. We have chosen baselines that are typical and widely recognized within the community, acknowledging that the investigation of other promising retrievers remains a task for future research. Our focus in this work is strictly on the application of retrievers for function annotation tasks. However, it is crucial to consider other downstream applications in future studies. For instance, protein engineering tasks, where the goal is to annotate proteins with minor sequence variations, present an important area for application. Another limitation of our current approach is the exclusive use of the ProtIR framework with the encoder, without integrating protein language models, primarily to minimize computational expenses. Exploring the integration of this framework with larger models could yield significant insights and advancements in the field.

## B    PSEUDO-CODE FOR PROTIR

The pseudo-code of ProtIR is shown in Alg. 1.

---

**Algorithm 1** EM Iterative Refinement Algorithm

---

**Input:** Labeled proteins $\boldsymbol{x}_L$ and their function labels $\boldsymbol{y}_L$, unlabeled proteins $\boldsymbol{x}_U$.
**Output:** Function labels $\boldsymbol{y}_U$ for unlabeled proteins $\boldsymbol{x}_U$.

1: Pre-train $q_\psi$ with $\boldsymbol{y}_L$ according to (1);
2: Pre-train $p_\phi$ on fold classification according to (3);
3: **while** not converge **do**
4:     ⊡ *E-step: Predictor Learning*
5:     Annotate unlabeled proteins with $p_\phi$ and $\boldsymbol{y}_L$ according to (2);
6:     Update $q_\psi$ with (10);
7:     ⊡ *M-step: Retriever Learning*
8:     Annotate unlabeled proteins with $q_\psi$;
9:     Denote the sampled labeled as $\hat{\boldsymbol{y}}_U$;
10:     Set $\hat{\boldsymbol{y}} = (\boldsymbol{y}_L, \hat{\boldsymbol{y}}_U)$ and update $p_\phi$ with (12);
11: **end while**
12: Classify each unlabeled protein $\boldsymbol{x}_n$ with $p_\psi$ and $q_\phi$

---

## C   DATASET DETAILS

Table 4: Dataset statistics.

| Dataset | # Train | # Validation | # Proteins # 30% Test / # 50% Test / # 95% Test |
|---|---|---|---|
| Enzyme Commission | 15,550 | 1,729 | 720 / 1,117 / 1,919 |
| Gene Ontology | 29,898 | 3,322 | 1,717 / 2,199 / 3,416 |
| Fold Classification | 12,312 | - | - |

Dataset statistics are summarized in Table 4. Details are introduced as follows.

For evaluation, we adopt two standard function annotation tasks as in previous works (Gligorijević et al., 2021; Zhang et al., 2023a). The first task, Enzyme Commission (EC) number prediction, involves forecasting the EC numbers for proteins, categorizing their role in catalyzing biochemical reactions. We have focused on the third and fourth levels of the EC hierarchy (Webb et al., 1992), forming 538 distinct binary classification challenges. The second task, Gene Ontology (GO) term prediction, targets the identification of protein associations with specific GO terms. We select GO terms that have a training sample size between 50 and 5000. These terms are part of a classification that organizes proteins into functionally related groups within three ontological categories: molecular function (MF), biological process (BP), and cellular component (CC).

To construct a non-redundant dataset, all PDB chains are clustered, setting a 95% sequence identity threshold. From each cluster, a representative PDB chain is chosen based on two criteria: annotation presence (at least one GO term from any of the three ontologies) and high-quality structural resolution. The non-redundant sets are divided into training, validation and test sets with approximate ratios 80/10/10%. The test set exclusively contains experimentally verified PDB structures and annotations. We ensure that these PDB chains exhibit a varied sequence identity spectrum relative to the training set, specifically at 30%, 50%, and 95% sequence identity levels. Moreover, each PDB chain in the test set is guaranteed to have at least one experimentally validated GO term from each GO category.

For pre-training a protein structure retriever, we adopt the fold classfication task (Hou et al., 2018), which holds significant relevance in analyzing the relationship between protein structure and function, as well as in the exploration of protein evolution (Hou et al., 2018). This classification groups proteins based on the similarity of their secondary structures, their spatial orientations, and the sequence of their connections. The task requires predicting the fold class to which a given protein belongs.

For the training of our model, we utilize the main dataset obtained from the SCOP 1.75 database, which includes genetically distinct domain sequence subsets that share less than 95% identity, updated in 2009 (Murzin et al., 1995). This dataset encompasses 12,312 proteins sorted into 1,195 unique folds. The distribution of proteins across these folds is highly skewed: about 5% of the folds (61 out of 1,195) contain more than 50 proteins each; 26% (314 out of 1,195) have between 6 to 50 proteins each; and the majority, 69% (820 out of 1,195), consist of 5 or fewer proteins per fold. The sequence

lengths of the proteins in these folds vary, ranging from 9 to 1,419 residues, with most falling within the 9 to 600 range.

# D    IMPLEMENTATION DETAILS

In this subsection, we describe implementation details of retriever-based baselines and our methods.

**BLAST and PSI-BLAST.**    We obtained the BLAST+ Version 2.14.0 (Altschul et al., 1990; Camacho et al., 2009) as its command line application to retrieve similar sequences for proteins in test set. For each task, we firstly built a BLAST database using `-dtype prot` (indicating "protein" sequences) for the training sequences. We then searched against the database for similar sequences using `blastp` and `psiblast` to query with `-evalue 10` (default). The alignment score is used to rank the retrieved proteins.

**MMSeqs.**    We ran the MMSeqs2 (Steinegger & Söding, 2017) as another sequence-based retriever.    The sequence database was built for both training set and test set using `mmseqs createdb` command and the alignment results were obtained by searching the test database against the training database with `mmseqs search` with the default configuration: `-s 5.7 -e 0.001 --max-seqs 300`. Finally, the alignment results were converted into readable table using the `mmseqs convertalis` and the (alignment) bit score was used to rank the retrieved records.

**TMAlign.**    TM-align (Zhang & Skolnick, 2005) is a pairwise structure alignment tools for proteins based on TM-score. TM-score is a normalized similarity score in $(0, 1]$ and can be used to rank the retrieved results. We ran the TM-align by enumerating all pairs between test set and training set, which forms a complete bipartite graph. Due to the intensive computational overhead, we executed the alignment with the flag `-fast` and then rank the results using TM-score.

**Foldseek.**    Foldseek (van Kempen et al., 2023) is run to obtain structure-based retrieved results. We created a Foldseek database for all structures in the training set using `foldseek createdb` and created search index with `foldseek createindex`. Then we searched for each structure in test set against the training database using command `foldseek easy-search`. All commands above were executed using 3Di+AA Gotoh-Smith-Waterman (`--alignment-type 2`) with the default parameters: `-s 9.5 --max-seqs 1000 -e 0.001 -c 0.0` and the alignment bit scores are used for ranking.

**Progres.**    Progres (Greener & Jamali, 2022) is a structure-based protein retrieval method based on a neural graph encoder. Firstly, we downloaded the code from the official repository as well as the trained model weights. Then we computed the graph embeddings for all the protein structures in both training and test set and all-vs-all pairwise similarity scores between them. The similarity score, as defined by Greener & Jamali (2022), is a normalized version of cosine similarity or formally $(\boldsymbol{v}_1 \cdot \boldsymbol{v}_2 / \|\boldsymbol{v}_1\| \|\boldsymbol{v}_2\| + 1)/2$. The similarity scores are used for ranking.

**TM-Vec.**    TM-vec (Hamamsy et al., 2022) is a neural sequence alignments tool that leverages structure-base similarity data in protein databases for training. To search the retrieved results between test and training set, we downloaded and ran the codes from its official repository. Specifically, we downloaded the pretrained weights for encoders named as `tm_vec_cath_model_large.ckpt`. We then built up the search database for the protein sequences in training set by running `tmvec-build-database` and `build-fasta-index` with default parameters.    Finally, the search was performed against the database above by setting query as test set with `--k-nearest-neighbors 10`. The predicted TM-score from the model is used for ranking.

For all retriever-based methods, we choose the top-$\{1, 3, 5, 10\}$ similar proteins from the training set and tune the temperature $\tau \in \{0.03, 0.1, 1, 10, 100\}$ according to the performance on validation sets. For neural methods, we use a batch size of $8$ and an SGD optimizer with learning rate 1e-3, weight decay 5e-4 and momentum $0.9$ for training. The models will be trained for 500 epochs and the learning rate will decay to one tenth at the 300-th and 400-th epoch. Other training details have been introduced in Sec. 5.

# E    ADDITIONAL EXPERIMENTS

## E.1    APPLYING RETRIEVER TO REAL-WORLD FUNCTION ANNOTATION

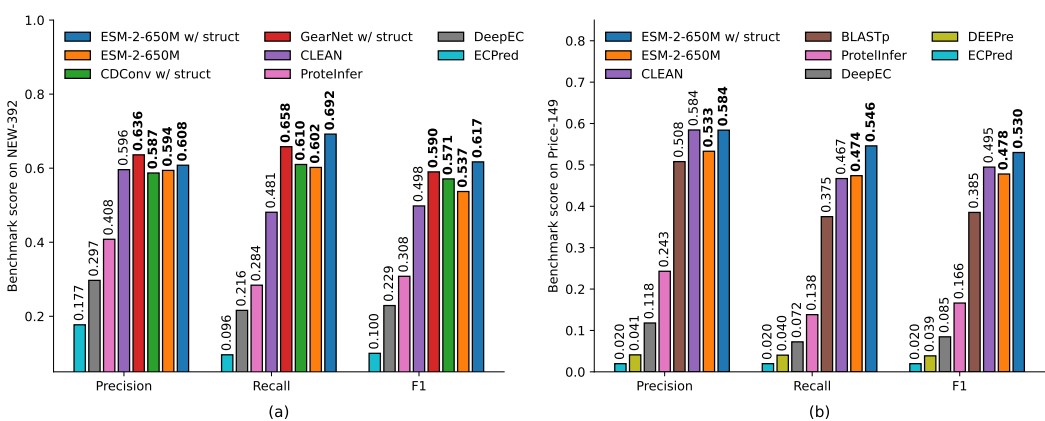

Figure 3: Quantitative comparison of the proposed retrievers with other EC number prediction tools on NEW-392 and Price-149 test sets. Results of our four neural retrievers are highlighted as bold.

In addition to the benchmark results presented in Sec. 5.2, we now extend to studies that explore EC number prediction under more real-world and challenging settings (Yu et al., 2023; Sanderson et al., 2021). Specifically, CLEAN (Yu et al., 2023) introduces a contrastive supervised learning approach that aligns protein representations with analogous enzyme commission numbers, an approach which has been substantiated through empirical validation. In this work, we deploy our proposed retrievers on their test sets without any fine-tuning on their respective training sets. This methodological choice is made to demonstrate the effectiveness of our retrieval-based approach in realistic settings.

**Setup.** We closely follow CLEAN (Yu et al., 2023) settings for evaluation. Baselines are trained on or retrieved against the collected Swiss-Prot dataset in Yu et al. (2023) with 227,363 protein sequences. Two independent test sets are used for a fair and rigorous benchmark study. The first, an enzyme sequence dataset, includes 392 sequences that span 177 different EC numbers. These sequences were released post-April 2022, subsequent to the proteins in our training set, reflecting a real-world scenario where the Swiss-Prot database serves as the labeled knowledge base, and the functions of the query sequences remain unidentified. The second test set, known as Price-149, consists of experimentally validated findings detailed by Price et al. (2018). This dataset, initially curated by Sanderson et al. (2021) as a benchmark for challenge, features sequences that were previously mislabeled or inconsistently annotated in automated systems.

**Methods.** We select six EC number prediction tools as baselines: CLEAN (Yu et al., 2023), ProteInfer (Sanderson et al., 2021), BLASTp (Altschul et al., 1990), ECPred (Dalkiran et al., 2018), DeepEC (Ryu et al., 2019), DEEPre (Li et al., 2018), the results of which are directly taken from the CLEAN paper (Yu et al., 2023). For comparison, we test the peformance of four neural retrievers presented in our paper : GearNet *w/ struct*, CDConv *w/ struct*, ESM-2-650M, ESM-2-650M *w/ struct*. Due to the large size of Swiss-Prot training set, we do not consider predictor-based methods and the ProtIR framework that requires training. This decision allows for a focused comparison on the effectiveness of retrieval-based approaches.

It is important to note that structure-based retrievers, such as GearNet and CDConv, require protein structures for input, which are not experimentally available for most proteins in Swiss-Prot. However, with the advent of the AlphaFold Database (Varadi et al., 2022), accurate structure predictions for the Swiss-Prot proteins made by AlphaFold2 are now accessible. For the purposes of our model, we search the available structures directly from the AlphaFold Database, successfully retrieving structures for 224,515 out of 227,363 proteins in the training (retrieved) set. A similar approach was adopted for the NEW-392 test set, from which structures for 384 proteins were obtained. In the case of the Price-149 dataset, the lack of UniProt IDs complicates the retrieval of corresponding structures from the AlphaFold Database. Additionally, running AlphaFold2 predictions for these proteins would

Table 5: $F_{max}$ on sequence-based tasks with predictors and retrievers based on PLMs.

| | Method | Subloc. | Binloc. | Sol. | Beta. | Fluoresence | Stability | AAV | GB1 | Thermo. |
|---|---|---|---|---|---|---|---|---|---|---|
| | | Acc. | Acc. | Acc. | Spearmanr | Spearmanr | Spearmanr | Spearmanr | Spearmanr | Spearmanr |
| **P** | ESM-2-650M | **82.5** | **92.5** | **74.7** | **0.898** | **0.680** | **0.695** | **0.800** | **0.678** | **0.645** |
| **R** | ESM-2-650M | 67.8 | 83.6 | 55.7 | 0.778 | 0.354 | 0.587 | 0.311 | 0.356 | 0.561 |
| | ESM-2-650M *w/ struct.* | 68.9 | 84.6 | 51.6 | 0.805 | 0.438 | 0.498 | 0.447 | 0.490 | 0.603 |

[*] **P**: predictor-based; **R**: retriever-based; **bold**: the best results.

be a time-consuming process. Therefore, we have chosen to exclude the two structure-based retriever baselines from our evaluation of the Price-149 test set.

**Results.** The results are plotted in Fig. 3. Here is our analysis of the findings:

First, it is evident that all four of our retrievers surpass the performance of CLEAN on the NEW-392 test set in F1 score, despite not undergoing any supervised training on the training set—a process that CLEAN underwent. This underscores the potency of retriever-based approaches.

Second, despite the lack of experimentally determined structures, neural structure retrievers demonstrate high performance with AlphaFold2-predicted structures, as shown in Fig. 3(a). Here, GearNet retrievers exhibit superior performance over the supervised retriever CLEAN and the PLM-based retriever ESM-2-650M. This exemplifies the data efficiency of structure-based retrieval methods in function determination, circumventing the need for large-scale training datasets.

Furthermore, the strategy of integrating structural data into PLM-based retrievers proves to be effective for EC number prediction, with observable enhancements on both test sets. Specifically, on the more challenging Price-149 set, while CLEAN slightly outperforms ESM-2-650M, it falls short against ESM-2-650M when structural information is incorporated. This reaffirms the significance of structural similarity in function similarity assessments.

To conclude, retriever-based methods continue to demonstrate their potential in practical scenarios, emphasizing the critical role of modeling similarities between proteins.

### E.2 RESULTS ON SEQUENCE-BASED PROPERTY PREDICTION TASKS

In addition to the experiments discussed in Sec. 5.4, where we evaluate PLM-based retrievers using only EC and GO, we extend our evaluation to test the ESM-2-650M model on a broader range of sequence-based function prediction tasks. We select nine tasks from the PEER benchmark (Xu et al., 2022), including GB1 fitness (Dallago et al., 2021), AAV fitness (Dallago et al., 2021), Thermostability (Dallago et al., 2021), Fluorescence (Sarkisyan et al., 2016), Stability (Rocklin *et al.*, 2017), beta-lactamase activity (Gray et al., 2018), Solubility (Khurana et al., 2018), Subcellular localization (Almagro Armenteros et al., 2017), and Binary localization (Almagro Armenteros et al., 2017). Following the default dataset split in the PEER benchmark, we employ ESM-2-650M as a predictor, retriever, and retriever with structural information.

Our results, present in Table 5, reveal consistent benefits in injecting structural information into protein language models to enhance retriever performance, even when the datasets lack protein structures. However, we note a notable performance gap between retriever-based methods and predictor-based methods for these sequence-based tasks. This discrepancy may stem from the limited diversity in the training set, where the considered protein engineering tasks primarily involve sequences with only one or two mutations, making it challenging to generalize to high-order mutants using simple retriever-based methods. Future research should focus on refining retriever-based approaches to surpass predictor-based methods on these sequence-based benchmarks, potentially requiring further exploration and enhancements.

### E.3 ANALYSIS ON PROTIR FRAMEWORK

#### E.3.1 COMPARISON WITH ENSEMBLE BASELINE

To demonstrate the efficacy of the ProtIR framework, we conducted a comparison involving the ProtIR-augmented GearNet and CDConv predictors against a basic ensemble baseline. This ensemble

Table 6: $F_{max}$ on EC and GO prediction with iterative refinement and ensembling baselines.

| Model | Method | EC | | | GO-BP | | | GO-MF | | | GO-CC | | |
|---|---|---|---|---|---|---|---|---|---|---|---|---|---|
| | | 30% | 50% | 95% | 30% | 50% | 95% | 30% | 50% | 95% | 30% | 50% | 95% |
| GearNet | Predictor | 0.700 | 0.769 | 0.854 | 0.348 | 0.359 | 0.406 | 0.482 | 0.525 | 0.613 | 0.407 | 0.418 | 0.458 |
| | Retriever | 0.671 | 0.744 | 0.822 | 0.391 | 0.419 | 0.482 | 0.497 | 0.548 | 0.626 | 0.377 | 0.387 | 0.434 |
| | Ensemble | 0.720 | 0.797 | 0.861 | 0.394 | 0.421 | 0.486 | 0.512 | 0.551 | 0.630 | 0.423 | 0.437 | 0.464 |
| | **ProtIR** | 0.743 | 0.810 | 0.881 | 0.409 | 0.431 | 0.488 | 0.518 | 0.564 | 0.650 | 0.439 | 0.452 | 0.501 |
| CDConv | Predictor | 0.634 | 0.702 | 0.820 | 0.381 | 0.401 | 0.453 | 0.533 | 0.577 | 0.654 | 0.428 | 0.440 | 0.479 |
| | Retriever | 0.719 | 0.784 | 0.843 | 0.409 | 0.434 | 0.494 | 0.536 | 0.584 | 0.661 | 0.387 | 0.397 | 0.438 |
| | Ensemble | 0.724 | 0.802 | 0.864 | 0.414 | 0.438 | 0.495 | 0.555 | 0.596 | 0.665 | 0.431 | 0.443 | 0.478 |
| | **ProtIR** | 0.769 | 0.820 | 0.885 | 0.434 | 0.453 | 0.503 | 0.567 | 0.608 | 0.678 | 0.447 | 0.460 | 0.499 |
| PromptProtein | | 0.765 | 0.823 | 0.888 | 0.439 | 0.453 | 0.495 | 0.577 | 0.600 | 0.677 | 0.532 | 0.533 | 0.551 |

Table 7: Training time comparison of different methods.

| Model | Pre-training Time | Pre-training Dataset | Fine-tuning Time | Total Time |
|---|---|---|---|---|
| **ESM-2-650M predictor** | >1K GPU hours | 60M sequences (UniRef50) | 50 GPU hours | >1K GPU hours |
| **ESM-2-650M retriever** | >1K GPU hours | 60M sequences (UniRef50) | - | >1K GPU hours |
| **GearNet predictor** | - | - | 6 GPU hours | 6 GPU hours |
| **GearNet retriever** | 6 GPU hours | 10K structures (SCOPe) | - | 6 GPU hours |
| **GearNet ProtIR** | 12 GPU hours | 10K structures (SCOPe) 10K-20K structures (EC or GO) | 4 GPU hours | 16 GPU hours |

approach involves averaging the predictions made by the predictor and its corresponding retriever, with the results presented in Table 6.

The results in the table reveal that while ensembling serves as a robust baseline for most tasks, our method is able to consistently enhance this baseline, achieving an improvement in the range of approximately 2% to 4% in terms of $F_{max}$. This improvement highlights the added value and effectiveness of the ProtIR framework in enhancing prediction accuracy across various tasks.

### E.3.2   TIME ANALYSIS

To evaluate the efficiency of the ProtIR framework, we list the training times for various function annotation methods, both with and without protein language models (PLMs), in Table 7. Notably, since the inference time for all methods typically does not exceed 1 GPU hour, we exclude it from our comparison. The table indicates that PLM-based methods, such as ESM-2-650M, often require massive pre-training, involving thousands of hours on millions of protein sequences. In contrast, structure-based methods utilizing the ProtIR framework can attain comparable performance levels without such time-consuming pre-training phases. These methods, by merely pre-training on datasets of the order of tens of thousands and applying iterative refinement in downstream tasks, demonstrate competitive performance when compare against PLM-based approaches. This finding underscores the efficiency and effectiveness of structure-based methods and the ProtIR framework in the realm of protein function annotation.

### E.4   HYPERPARAMETER CONFIGURATION

**Hyperparameter analysis for retriever-based methods.** To investigate the impact of the number of retrieved neighbors ($k$) and the temperature parameter ($\tau$) on the performance of function annotation in retriever-based methods, we plot a heatmap for this hyperparameter analysis, as shown in Fig. 4. We observe that a temperature of $\tau = 0.03$ yields the most effective results for scaling the cosine similarity between protein representations. This optimal setting can be attributed to the nature of cosine similarity, which ranges between $[-1, 1]$; without amplification by the temperature, there is minimal variation in the weights assigned to different proteins.

Furthermore, we note that at lower values of $k$, the effect of the temperature parameter is relatively minor, primarily because most of the retrieved proteins tend to have the same EC number. However, as $k$ increases, leading to a wider variety of retrieved EC numbers, the temperature becomes more

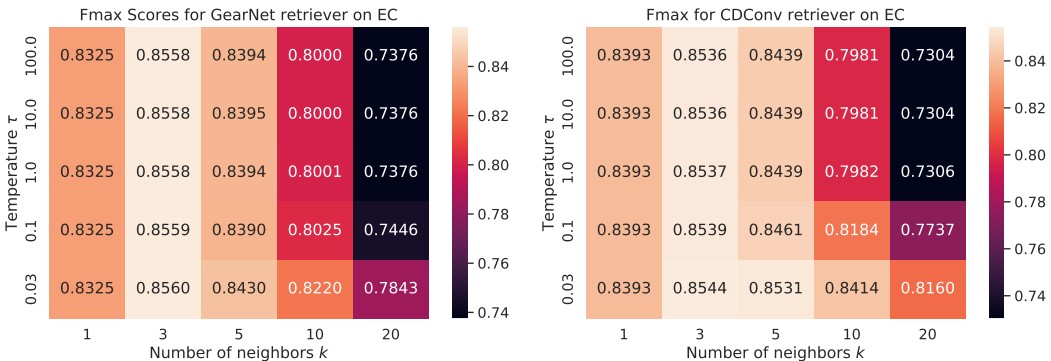

Figure 4: Change of $F_{max}$ on EC with respect to $k$ and $\tau$ in Eqs.(2)(4) for two retrievers.

influential. In such scenarios, it serves to emphasize proteins that are more similar to the query protein, thereby refining the function annotation process. This understanding highlights the importance of carefully selecting the values of $k$ and $\tau$ to optimize the performance of retriever-based methods.

**Hyperparameter tuning for ProtIR framework.** The tuning process for the ProtIR framework is divided into two main stages: the pre-training stage and the refinement stage.

In the pre-training stage, for both predictors and retrievers, we adhere to the optimal hyperparameters established in prior research (Zhang et al., 2023a). This includes settings for the learning rate, batch size, and the number of epochs. The model that achieves the best performance on the validation set is then selected to proceed to the refinement stage.

During the refinement stage, the predictor and retriever are iteratively refined. In each iteration, it is crucial to balance the models' convergence with the goal of fitting pseudo-labels, while also being mindful of potential overfitting. To maintain this balance, we closely monitor performance metrics on the validation set and halt training when no further improvements are observed. Notably, test set performance is not considered during training to ensure a fair comparison.

Based on our experience, training for approximately 30 epochs during both the E-step and M-step is typically sufficient for the convergence of both the predictor and retriever. Moreover, the validation performance often stabilizes after around five rounds. The final step involves selecting the model with the best performance on the validation set and subsequently evaluating it on the test set.

