# OpenReview forum: "Illuminating Protein Function Prediction through Inter-Protein Similarity Modeling"
_ICLR.cc/2024/Conference — Submitted to ICLR 2024_

### Official Review · Reviewer_v1LN · 2023-10-31

**Soundness:** 2 fair
**Presentation:** 3 good
**Contribution:** 2 fair
**Rating:** 6
**Confidence:** 3

**Summary:**

This paper studies protein function prediction by combining 2 established methods. The first being "retriever based methods" which retrieve proteins similar to a query protein and infer similar protein function, the second is "preditcion" methods that learn to predict the protein function directly from labelled data.
Retriever based models produce a representation for each protein that can be used to generate a similarity score between two proteins.
Function prediction models output a binary vector where each dimension represents a particular functional property. Function prediction models are (initially) trained on limited labelled data.

They use a training scheme that refines both methods in an iterative EM process. In several rounds they first fix the retriever model and use it to predict labels to refine the function predictor model (their E step), then the fix the function predictor model and finetune the retriever model using labels from the predictor as "pseudo labels" (the M step).
Several rounds of the EM process produces both prediction and retriever models that have better accuracy that the original model by a significant margin in their evaluation.
It is the use of labels for function prediction that are expected to help finetune the similarity (retriever) model.

**Strengths:**

The iterative EM training approach is reasonable, a form of distillation (or label propagation between two models)
Using the strengths of both models to enhance each other works.
It is a well written and thorough read with a "good sized" evaluation.
The reported results are significant improvements.

**Weaknesses:**

I would worry about robustness to changes in hyperparameters for the EM finetuning process (eg. epoch size learning rate etc.).
The method does depend on the idea that both pretrained models are able to give sufficiently accurate supervision to the other not to misguide, and that they have complimentary and transferable abilities. But in this case it does work, and so one could say the method works.

**Questions:**

How much do you think this method (iterative refinement between similarity and classification models) do you think is general? What other kinds of data might it extend to? Are there similar examples in other domains?

---

> ### Author Response · Authors · 2023-11-21
>
> Thanks for your insightful comments and golden suggestions! We respond to your concerns as below:
>
> >**Q1: I would worry about robustness to changes in hyperparameters for the EM finetuning process (eg. epoch size learning rate etc.).**
>
> Thanks for the question! We’ve added a section to discuss the hyperparameter setting and tuning in App. E.4. For ProtIR, we can directly inherit the best hyperparameters from existing works during pre-training. During the refinement stage, we tune the hyperparameter on the validation set to avoid looking at the test performance during training. We need to ensure the model converge at each interaction and avoid overfitting. To maintain this balance, we closely monitor performance metrics on the validation set and halt training when no further improvements are observed.
>
>
> >**Q2: How much do you think this method (iterative refinement between similarity and classification models) do you think is general? What other kinds of data might it extend to? Are there similar examples in other domains?**
>
> The method is quite general, as it can be applied on texts, images or graphs. To the best of our knowledge, there is little existing work on other data formats. But there are some works using the idea of iterative refinement for mutual enhancement of LLM and GNN-based methods [1].
>
> [1] Zhao et al. “Learning on Large-scale Text-attributed Graphs via Variational Inference” ICLR (2023)

---

### Official Review · Reviewer_fLin · 2023-11-01

**Soundness:** 3 good
**Presentation:** 3 good
**Contribution:** 3 good
**Rating:** 6
**Confidence:** 3

**Summary:**

This manuscript presents a comparative study between deep learning predictors and retriever-based methods for protein function prediction. The authors have conducted a comprehensive benchmark to analyze the performance of each approach. By focusing on the role of inter-protein similarity modeling, the paper sheds light on the nuances of protein function prediction and attempts to bridge the gap between modern deep learning techniques and traditional retrieval tools.

**Strengths:**

- Empirical Results: The study reports strong empirical results, showcasing the effectiveness of both predictors and retriever-based methods in protein function prediction tasks.
- Clarity of Writing: The manuscript is well-written, presenting a clear and structured narrative that effectively communicates the research and findings to the reader.

**Weaknesses:**

- Lack of Innovation: The paper appears to be an incremental work that combines predictive models with retriever-based methods without introducing significant methodological advancements or innovations.

- Theoretical Contribution: The manuscript could be improved by providing a deeper theoretical understanding or insights into why certain methods perform better and under what conditions each approach is preferable.

**Questions:**

- Innovation Clarification: The authors should clearly articulate any novel contributions of their work, particularly if there are innovative aspects beyond the combination of predictive and retriever-based methods.

- Theoretical Insights: A section discussing the theoretical implications of the findings would enrich the paper. This could include a discussion on the conditions under which each method excels or fails.

- Broader Impact and Ethical Considerations: The manuscript would benefit from a dedicated section on the broader impact and ethical implications of the research, particularly in the context of bioinformatics and potential applications in drug discovery.

- Compared to structure prediction, what's the most significance of function prediction of proteins?

- Is it feasible to integrate the methodology proposed in the manuscript with AlphaFold 2 to create an end-to-end system for protein function prediction？ Such integration could potentially harness the strengths of AlphaFold 2 in structural prediction and the comparative insights from the manuscript to offer a comprehensive solution for predicting protein functions. This could pave the way for more accurate and holistic predictions, significantly benefiting fields such as drug discovery, disease understanding, and synthetic biology.

**Details Of Ethics Concerns:**

The manuscript currently lacks a thorough discussion of the broader impact and ethical implications of the work. Topics such as data privacy, potential misuse, and ethical considerations in drug discovery should be addressed. This is especially important given the sensitivity and potential societal impact of research in bioinformatics.

---

> ### Author Response · Authors · 2023-11-21
>
> Thanks for your insightful comments and golden suggestions! We respond to your concerns as below:
>
> >**Q1: Innovation Clarification: The authors should clearly articulate any novel contributions of their work, particularly if there are innovative aspects beyond the combination of predictive and retriever-based methods.**
>
> Thanks for the question! We add a paragraph in the Introduction and Related Work sections to discuss the contribution of this work.
>
> Specifically, we find that there is still a lack of systematic study of modern deep learning methods and retriever-based methods for function annotation. The most closely related work should be CLEAN [1], where they benchmark their proposed supervised contrastive learning methods with several retriever-based methods. However, there are still many deep learning baselines not considered in that paper. In this paper, we include more modern baselines in our benchmark and add another experiment on the CLEAN setup to show the advantage of our method in App. E.
>
> From method perspective, there are many existing works using retriever-based and transductive learning methods for other machine learning applications, but there has been little effort put in the protein function annotation task. One related work along this direction is a concurrent work - RSA [2], which studies other sequence-based property prediction tasks. Our work is among the first to study how to use retriever-based methods for enhancing predictor-based methods.
>
> [1] Yu, Tianhao, et al. "Enzyme function prediction using contrastive learning." Science 379.6639 (2023): 1358-1363.
>
> [2] Ma, Chang, et al. "Retrieved Sequence Augmentation for Protein Representation Learning." bioRxiv (2023): 2023-02.
>
>
> >**Q2: Theoretical Insights: A section discussing the theoretical implications of the findings would enrich the paper. This could include a discussion on the conditions under which each method excels or fails.**
>
> We want to argue that this paper is an application-oriented paper, where we put our focus on a practical function annotation setting. Our proposed method is principal as derived in Sec. 3 and achieves good performance on our benchmark. We believe the contribution of this paper is enough for a conference paper and we leave the theoretical analysis as future work.
>
> >**Q3: Broader Impact and Ethical Considerations: The manuscript would benefit from a dedicated section on the broader impact and ethical implications of the research, particularly in the context of bioinformatics and potential applications in drug discovery.**
>
> Thanks for the suggestion! We’ve added a paragraph to discuss the potential negative impact of the paper in the App. A during rebuttal.
>
> >**Q4: Compared to structure prediction, what's the most significance of function prediction of proteins?**
>
> Function prediction is clearly an important problem in the protein domain, as the ultimate goal in this domain is to design proteins with desired functions. This problem has been studied for decades [1,2,3]. Structure prediction is just an intermediate step for us to understand protein functions, as structure directly determines protein functions.
>
> [1] Ashburner et al. "Gene ontology: tool for the unification of biology. The Gene Ontology Consortium". Nature Genetics (2000). 25 (1): 25–9.
>
> [2] Webb et al. “Enzyme nomenclature 1992: recommendations of the Nomenclature Committee of the International Union of Biochemistry and Molecular Biology on the nomenclature and classification of enzymes.” Academic Press (1992)
>
> [3] Yu, Tianhao, et al. "Enzyme function prediction using contrastive learning." Science 379.6639 (2023): 1358-1363.
>
> >**Q5: Is it feasible to integrate the methodology proposed in the manuscript with AlphaFold 2 to create an end-to-end system for protein function prediction?**
>
> Thanks for proposing this idea. We’d like to argue that there is no need to combine with AlphaFold2 for an end-to-end system. First, AlphaFold2 focuses on the structure prediction problem, which has been almost solved and is not relevant to the problem we studied in this paper. We can directly take the experimentally determined or predicted structures as input to our model, instead of connecting with AlphaFold2. Second, using AlphaFold2 during training will be extremely time-consuming, which violates the goal of this paper to improve function annotation efficiency with little additional training.

---

> > ### Comment · Reviewer_fLin · 2023-11-21
> >
> > Thanks for your kindly response, my concerns are well resolved and I want to raise my score.

---

### Official Review · Reviewer_gh7e · 2023-11-01

**Soundness:** 2 fair
**Presentation:** 2 fair
**Contribution:** 2 fair
**Rating:** 6
**Confidence:** 3

**Summary:**

The paper evaluates and compares sequence- and structure-based methods against predictors for protein function annotation. It demonstrates that retriever methods perform similarly to predictor-based methods. Next, it introduces an iterative training strategy that alternates between a retriever and a predictor to enhance predictive performance. This approach shows improvements in prediction accuracy and efficiency when compared to other methods

**Strengths:**

The paper is well written and easy to read.

The application of EM iterative algorithm to the problem of function prediction seems interesting and to have potential.
The same can be said of the framework proposed ProtIR

**Weaknesses:**

The paper is not self-contained.

The paper lacks a dataset section. While the paper mentions the use of some datasets, they are never adequately described. For example, how many proteins are included, and where were they obtained from? The description of the tasks is also lacking; they are only briefly covered in section 5.1.

The evaluation procedure is not adequately described in the paper. There is no explanation of how the training and test sets are obtained, or what kind of evaluation is performed.

The paper claims that their approach reduces computational requirements while needing less running time, yet this is not supported by experiments or any comparisons to other methods. Additionally, it is difficult to properly judge the reported performance without knowing the size of the dataset used.

The paper uses inter-protein similarity to model and predict function, but it does not compare this approach to important and state-of-the-art methods that model inter-protein similarity. For instance:

* K. Wu, L. Wang, B. Liu, Y. Liu, Y. Wang and J. Li, "PSPGO: Cross-Species Heterogeneous Network Propagation for Protein Function Prediction," in IEEE/ACM Transactions on Computational Biology and Bioinformatics, vol. 20, no. 3, pp. 1713-1724, 1 May-June 2023, doi: 10.1109/TCBB.2022.3215257
* Torres, M., Yang, H., Romero, A.E. et al. Protein function prediction for newly sequenced organisms. Nat Mach Intell 3, 1050–1060 (2021). https://doi.org/10.1038/s42256-021-00419-7
* Shuwei Yao, Ronghui You, Shaojun Wang, Yi Xiong, Xiaodi Huang, Shanfeng Zhu, NetGO 2.0: improving large-scale protein function prediction with massive sequence, text, domain, family and network information, Nucleic Acids Research, Volume 49, Issue W1, 2 July 2021, Pages W469–W475, https://doi.org/10.1093/nar/gkab398


Minor comments:

It would also be interesting to see the performance in terms of s min, which is normally used to evaluate function prediction.

The authors assume that there is independence between the function labels. This assumption does not hold for protein function prediction using GO terms. The authors should better justify why they make this design decision.

**Questions:**

In section 2, authors claim that "A notable advantage of these neural retrievers over traditional methods is their flexibility in fine-tuning for specific functions, as will discuss in next section". Is not clear to me where this is discussed or pointed, could you please elaborate this further?

---

> ### Author Response · Authors · 2023-11-21
>
> Thanks for your insightful comments and golden suggestions! We respond to your concerns as below:
>
> >**Q1: The paper lacks a dataset section. The evaluation procedure is not adequately described in the paper.**
>
> Thanks for the questions. During rebuttal, we’ve added more dataset details and evaluation setup in App. C.
>
> >**Q2: The paper claims that their approach reduces computational requirements while needing less running time, yet this is not supported by experiments or any comparisons to other methods.**
>
> Thanks for the question. In the revised version, we include the time comparison of different methods in App. E.3.2.
>
> >**Q3: The paper uses inter-protein similarity to model and predict function, but it does not compare this approach to important and state-of-the-art methods that model inter-protein similarity.**
>
> Thanks for bringing our attention to these works. We’ve added a paragraph to discuss the relation between our work and this stream of works in App. A.
>
> To clarify, instead of focusing on inter-protein similarity, these works use protein-protein interaction networks for function annotation, such as STRING. These networks map both direct physical and indirect functional interactions among proteins. A key limitation of these methods is that they are not able to make predictions for newly sequenced proteins absent in existing PPI networks.
> Moreover, knowing protein-protein interactions is essentially a more difficult challenge, as it requires a more comprehensive understanding of protein properties.
> These problems make this line of work hard to use in real-world settings. Therefore, to ensure a fair comparison, we do not include these baselines in our benchmark.
>
> Instead, during rebuttal, we add CLEAN paper [1] and other corresponding baselines on function annotation for comparison, which is more related with our work. The results are shown in App. E.1, which clearly demonstrates the effectiveness of our methods.
>
>
> >**Q4: The authors assume that there is independence between the function labels. This assumption does not hold for protein function prediction using GO terms. The authors should better justify why they make this design decision.**
>
> This is a good point! We simply treat the function annotation problem as a multiple binary classification problem following common practice in machine learning. This is a reasonable assumption as most functions can be determined by the protein sequence and structure. **Also, we want to emphasize that the focus of this paper is to study general function annotation tasks without any assumption on
> the function-specific characteristics.** Nevertheless, we admit the effectiveness of modeling the tree structure of GO terms and leave this as future work.
>
> >**Q6: In section 2, authors claim that "A notable advantage of these neural retrievers over traditional methods is their flexibility in fine-tuning for specific functions, as will discuss in next section". Is not clear to me where this is discussed or pointed, could you please elaborate this further?**
>
> Thanks for the question! One advantage of neural retrievers is that they can be fine-tuned for specific functions using contrastive supervised learning, as done in the CLEAN paper [1]. This is exactly what to do to refine the retriever in the ProtIR framework.
>
> [1] Yu, Tianhao, et al. "Enzyme function prediction using contrastive learning." Science 379.6639 (2023): 1358-1363.

---

> > ### Comment · Reviewer_gh7e · 2023-11-22
> >
> > The authors answered most of my concerns; I will increase my score accordingly.

---

### Official Review · Reviewer_Fp4G · 2023-11-02

**Soundness:** 3 good
**Presentation:** 4 excellent
**Contribution:** 3 good
**Rating:** 6
**Confidence:** 4

**Summary:**

This paper begins by benchmarking various methods for protein function annotation (specifically, enzyme classification and GO term prediction).  The paper segregates methods into two categories, nearest-neighbor techniques ("retriever-based methods") and standard classification algorithms ("predictor-based methods").  It then introduces a new method, ProtIR, that uses an EM-like algorithm to alternate between retrieval and prediction.  Empirical results suggest that this approach improves upon the state of the art.

**Strengths:**

The paper addresses an important and longstanding problem, and it appears to provide a significant improvement in classification accuracy relative to the state of the art.

The ProtIR method is elegant and well described.

The ablation experiments help to convince me that the results are legitimate.

The proposed method is much cheaper than pretraining a large language model.

**Weaknesses:**

Please cite the original BLAST publication (Altschul 1990) rather than the 2004 paper cited here.  It is not correct to say that BLAST operates "under the assumption that proteins with similar sequences likely posess similar functions."  BLAST has nothing to do with functional inference, per se.  It identifies evolutionary relationships.

I found retriever/predictor terminology confusing at first.  Please define these terms explicitly.

I was concerned by the baseline method proposed in Section 2.2.  BLAST is a fast heuristic approximation of the Smith-Waterman DP algorithm.  But even SW is not as good at detecting homologies as more advanced methods that rely on profile-profile alignment (e.g., HHPred/HHSearch).  Indeed, even PSI-BLAST (which is part of the BLAST package) is better than BLAST.  So I can't help but wonder why BLAST is being used here.  It's also not at all clear to me that Equation 2 represents a smart way to use BLAST. Indeed, one of BLAST's strengths is its statistical confidence estimation procedure.  As such, treating the BLAST output as an uncalibrated score and then doing ad hoc normalization seems like a bad idea.

The related work is almost comically dense, as is unfortunately required by this conference format.  But the upshot is that it's very difficult to get a sense for where you think ProtIR falls in terms of novelty relative to all of these prior methods.  The final sentences of the second and third paragraphs are the only hints along these lines.  Rather than a laundry list of existing methods, I'd prefer that you cite a couple of reviews and then use the space to explain how your method relates to the state of the art.

I found it striking that almost all of the related work is from the last few years.  In fact, people have been working hard on this problem for decades.  See, e.g., Melvin PLOS CB 2011 for a retriever-based method that operates on protein embeddings using sequence and structure.  Or see work by JP Vert on kernels for protein sequence similarity.  This review might be a good place to get a sense for the long history in this general area: https://academic.oup.com/bib/article/19/2/231/2562645

Minor: typo "orignial": "v.s." -> "vs."

**Questions:**

What are the two or three most closely related prior methods for solving this problem, and how does your method differ from them?

Have you explored using alternatives to BLAST as your retriever?

Why did you use the alignment score rather than the E-value from BLAST?

---

> ### Author Response · Authors · 2023-11-21
>
> Thanks for your insightful comments and golden suggestions! We respond to your concerns as below:
>
> >**Q1: Please cite the original BLAST publication (Altschul 1990) rather than the 2004 paper cited here. BLAST has nothing to do with functional inference, per se. It identifies evolutionary relationships.**
>
> Thanks for the suggestion! We’ve followed your suggestion to replace the citation of BLAST to the original paper.
>
> >**Q2: I found retriever/predictor terminology confusing at first. Please define these terms explicitly.**
>
> Thanks for pointing this out! We’ve added a definition in Sec. 2.1 and 2.2 in the revised version.
>
> >**Q3: Why BLAST is used here instead of PSI-BLAST? Have you explored using alternatives to BLAST as your retriever? Why did you use the alignment score rather than the E-value from BLAST?**
>
> Table A: $F_\max$ for the performance of three sequence retrievers on EC and GO. (#/#/#) shows the results on the split (30%/50%/95%), respectively.
> |#Method|EC|GO-BP|GO-MF|GO-CC|
> |:----:|:----:|:----:|:----:|:----:|
> | BLAST | 0.740 / 0.806 / 0.872 | 0.344 / 0.373 / 0.448 | 0.505 / 0.557 / 0.640 | 0.275 / 0.284 / 0.347 |
> | BLAST (e-value) | 0.741 / 0.814 / 0.874 | 0.382 / 0.411 / 0.467 | 0.473 / 0.528 / 0.609 | 0.371 / 0.390 / 0.430 |\\
> | PSI-BLAST | 0.642 / 0.705 / 0.798 | 0.341 / 0.364 / 0.442 | 0.433 / 0.482 / 0.573 | 0.354 / 0.365 / 0.420 |
>
> Thanks for the suggestion! We’ve follow your suggestions to include both baselines during rebuttal, the results of which are shown in Table A.
>
> **About PSI-BLAST:** As shown in the table, PSI-BLAST does not perform better than BLAST on most tasks. Here is our explanation. PSI-BLAST, or position-specific iterated BLAST, is equivalent to running one round of BLASTp and use the resulting MSA to build a profile and thus position-specific score matrix (PSSM) to imply conservation patterns. The PSSM can guide another round of search, which can be repeated iteratively. However, it requires searching against large sequence database to build an effective PSSM, which performs badly in our settings since our retrieval database is simply the training set of ~20k sequences. The same rationale also applies to HHsearch/HHpred, and we note that these methods are good for searching high-quality MSA but not a good kernel for retrieval w.r.t. a limited database.
>
> **About e-value:** We’d like to emphasize that e-value and alignment scores have strong correlation, which may result in the similar set of retrieved proteins. The minor difference here is how to use these scores for weighting different samples. For e-value, we need to take the logarithm and then tune the temperature to rescale the similarity. As shown in the table, there is no obvious conclusion whether e-value or alignment scores are better for function annotation.
>
> >**Q4: (1) The related work is almost comically dense, as is unfortunately required by this conference format. Rather than a laundry list of existing methods, I'd prefer that you cite a couple of reviews and then use the space to explain how your method relates to the state of the art. (2) Almost all of the related work is from the last few years. In fact, people have been working hard on this problem for decades. (3) What are the two or three most closely related prior methods for solving this problem, and how does your method differ from them?**
>
> Thanks for the suggestion! We’ve followed your suggestions to reorganize the related work section and added more related work in App. A. We also discuss the contribution of this work in the end of Introduce and Related Work section.
>
> We find that there is still a lack of systematic study of modern deep learning methods and retriever-based methods for function annotation. The most closely related work should be CLEAN [1], where they benchmark their proposed supervised contrastive learning methods with several retriever-based methods. However, there are still many deep learning baselines not considered in that paper. In this paper, we include more modern baselines in our benchmark and add another experiment on the CLEAN setup to show the advantage of our method in App. E.
>
> From method perspective, there are many existing works using retriever-based and transductive learning methods for other machine learning applications, but there has been little effort put in the protein function annotation task. One related work along this direction is a concurrent work - RSA [2], which studies other sequence-based property prediction tasks. Our work is among the first to study how to use retriever-based methods for enhancing predictor-based methods.
>
> [1] Yu, Tianhao, et al. "Enzyme function prediction using contrastive learning." Science 379.6639 (2023): 1358-1363.
>
> [2] Ma, Chang, et al. "Retrieved Sequence Augmentation for Protein Representation Learning." bioRxiv (2023): 2023-02.

---

> > ### Comment · Reviewer_Fp4G · 2023-11-22
> >
> > The authors have addressed my concerns in a satisfactory fashion.  I still am not convinced that the normalization approach is the right one, nor that PSI-BLAST is the best baseline.  It seems like more familiarity with the deep literature in this area is still in order.  I am leaving my score as is.

---

### Official Review · Reviewer_MNvs · 2023-11-02

**Soundness:** 3 good
**Presentation:** 2 fair
**Contribution:** 2 fair
**Rating:** 5
**Confidence:** 4

**Summary:**

In this paper, the authors focus on protein function prediction problem, and propose a variational pseudo-likelihood framework ProtIR, which conducts iterative refinement between predictors and retrievers to improve protein function prediction performance. It utilizes EM algorithm to optimize the function predictors and retrievers to integrate the advantages of the two models. The experimental results prove the effectiveness of the proposed method.

**Strengths:**

1. This paper comprehensively discusses the performance of methods based on predictors and retrievers, and proposes a novel iterative refinement framework to integrate the two models.
2. The proposed method achieves better performance compared to predictor-based methods and
 improves efficiency compared to protein language model-based methods.
3. This paper conducts comprehensive experiments to prove the performance and efficiency of the proposed method.

**Weaknesses:**

1. In the experimental section, the authors mainly present the experimental results, but the analysis is not sufficient.
2. In the method section, the definitions of some symbols in this paper are not clear, and in the experiments section, the description of the datasets is not sufficient.
3. The authors emphasize the improvement of computational efficiency, but in the experiment, the authors only provided a rough explanation and did not provide specific improvement results.

**Questions:**

1. In the first paragraph of the section 2.1, how is the representation of protein x defined? How are the sequence and structural information of proteins integrated?
2. In the last paragraph of the section 2.3, there was no analysis of the setting of parameter τ. How the different settings of this parameter will affect performance?
3. In the section 3.3 on page 4, what is the meaning of const in the formula? The authors did not provide an explanation.
4. In the experiments section, have the results in Tables 1, 2, and 3 been verified for statistical significance?
5. The authors should improve the standardization of the paper. For example: (1) In the section 3.3, the label of the first formula is missing. (2) In Figure 2 of the section 5.3, the labels for the axes are missing.

---

> ### Author Response · Authors · 2023-11-21
>
> Thanks for your insightful comments and golden suggestions! We respond to your concerns as below:
>
> >**Q1: In the experimental section, the authors mainly present the experimental results, but the analysis is not sufficient.**
>
> Thanks for the suggestion! We’ve revised the experiment section to add more analysis about the benchmark results. Please refer to Sec. 5.2.
>
> >**Q2: In the experiments section, the description of the datasets is not sufficient.**
>
> We’ve followed your suggestion to add a section for detailed dataset description in App. C during rebuttal.
>
> >**Q3: The authors emphasize the improvement of computational efficiency, but in the experiment, the authors only provided a rough explanation and did not provide specific improvement results..**
>
> Sorry for the confusion. We’ve added a section to discuss the training time of different methods in App. E.3.2 to show the efficiency of proposed methods.
>
> >**Q4: In the first paragraph of the section 2.1, how is the representation of protein x defined? How are the sequence and structural information of proteins integrated?**
>
> We use $x$ to denote a protein instead of its representations. To obtain its representation, we will feed its sequence and structure to any protein encoder, e.g., ESM-2, GearNet or CDConv. We’ve simplified the notation in the revised version.
>
> >**Q5: In the last paragraph of the section 2.3, there was no analysis of the setting of parameter τ. How the different settings of this parameter will affect performance?**
>
> Thanks for the question! We’ve added a new section discussing the effect of the temperature $\tau$ and the number of neighbors $k$ in App. E.4.
>
> >**Q6: In the section 3.3 on page 4, what is the meaning of const in the formula? The authors did not provide an explanation.**
>
> The const in the equation denotes the term irrelevant with the predictor $\psi$, which can be omitted in the E-step. We’ve added an explanation in the revised version.
>
> >**Q7: The authors should improve the standardization of the paper. For example: (1) In the section 3.3, the label of the first formula is missing. (2) In Figure 2 of the section 5.3, the labels for the axes are missing.**
>
> Thanks for pointing out these! We’ve fixed these in the revised version.

---

### Official Review · Reviewer_Rgn7 · 2023-11-03

**Soundness:** 3 good
**Presentation:** 2 fair
**Contribution:** 3 good
**Rating:** 5
**Confidence:** 3

**Summary:**

The paper studies protein function prediction, with a focus on the comparison between predictor-based and retriever-based methods. The authors introduce ProtIR, a transductive learning approach that iteratively refines an encoder and a predictor neural network. The experimental results first provide a comparison of a number of predictor-based and retriever-based approaches on EC and GO function prediction tasks. Then, they show that ProtIR results in an increased accuracy compared to the predictor-based baseline and three transductive learning baselines.

**Strengths:**

+ For the most part, the paper is well written and successful in motivating the research
+ The proposed approach is reasonable and mathematically described in sufficient detail
+ The results indicate that the proposed approach is beneficial
+ Benchmarking of numerous baseline approaches is a bonus in this paper

**Weaknesses:**

- unfortunately, the paper becomes difficult to follow when it comes to technical details.
-- Other than mentioning in one sentence in section 2.3 that encoder are pre-trained on the fold classification task with 16,712 proteins with 1,195 different folds, data sets are not specified anywhere in the paper. What data set you had available for EC and GO training and testing?
-- ProtIR iterates between the encoder model and prediction model fine-tuning steps. But, without knowing on what data this happens and what is the relationship between training and test data, it is difficult to imagine what is happening in the experiments. For example, is are fold classification and protein function data sets coming from different distributions? If so, what is the consequence on the experimental results?
-- the paper is vague when it comes to what neural networks are used as encoder model and predictor model. Only deep into the experimental sections it becomes clear that the same neural network (GerNet or CDConv) is used for both. But, does that mean that there are two copies of the neural net -- one serving as an encoder and another as a predictor?
-- the paper is inconsistent when it comes to explaining hyperparameters for the experiments
- using the structure encoders is a major limitation of this approach because it is only applicable on proteins with known structure. Proteins with known structure are a highly biased sample of the protein space and are typically well studied with a good understanding of their functional properties. This, the usefulness of proposed approach is limited

**Questions:**

The main weakness of the paper is that technical details about the implementation of the proposed method and the details about the data set and the experimental design are unclear. Without this, the readers need to hope that the code that was promised (upon acceptance of the paper) will be documented well enough to explain the missing details.

---

> ### Author Response · Authors · 2023-11-21
>
> Thanks for your careful reading! While some questions may be due to misunderstanding, we find that your suggestions are valuable and very helpful for us to improve the quality and clarity of our paper! We respond to your concerns below:
>
> >**Q1: What data set you had available for EC and GO training and testing?**
>
> Thanks for the question. In the revised version, we add more description about the datasets and splits in the App. C.
>
> >**Q2: Without knowing on what data this happens and what is the relationship between training and test data, it is difficult to imagine what is happening in the experiments. For example, is are fold classification and protein function data sets coming from different distributions? If so, what is the consequence on the experimental results?**
>
> Thanks for the suggestion. We did include the source of datasets in the original version and we’ve elucidated more about the dataset in the revised version in App. C. Here the fold classification dataset is collected from SCOPe, which is a different dataset with the Swiss-Prot dataset for function annotation.
>
> >**Q3: The paper is vague when it comes to what neural networks are used as encoder model and predictor model. Only deep into the experimental sections it becomes clear that the same neural network (GerNet or CDConv) is used for both. But, does that mean that there are two copies of the neural net -- one serving as an encoder and another as a predictor?**
>
> We want to clarify that it is not necessary to use the same neural network for both predictors and retrievers. Our framework is very flexible so that you can use arbitrary learnable predictors or retrievers. It is just our choice to use the same neural network (GearNet and CDConv) for predictors and retrievers in our experiments to make this simple. Also, the retriever and predictor do not share weights, as distinguished by the different notations $p_\phi$ and $q_\psi$.
>
> >**Q4: The paper is inconsistent when it comes to explaining hyperparameters for the experiments**
>
> We add more explanation about hyperparameter setting in App. E.4. Could you elaborate more about where the inconsistency you find in the hyperparameter setting?
>
> >**Q5: Using the structure encoders is a major limitation of this approach because it is only applicable on proteins with known structure. Proteins with known structure are a highly biased sample of the protein space and are typically well studied with a good understanding of their functional properties. This, the usefulness of proposed approach is limited.**
>
> We want to argue that relying on protein structures should not be a limitation of this work. With the advent of AlphaFold2, it is much easier and efficient to obtain accurate structures for most proteins. Also, there have been large amount of protein structures already released in AlphaFold Database, which can be easily searched for.
>
> To illustrate this, we test our structure encoders on the CLEAN datasets in App. E.1. For most proteins in the training (retrieved) sets (Swiss-Prot), we only have the protein sequences instead of structures. To solve the problem, we simply retrieve the predicted structures from AlphaFold Database according to the UniProt ID. Although there are a small portion of proteins without structures, our structure retriever can still outperform the CLEAN method on NEW-392 dataset. This shows that our method can work well on unseen proteins with AlphaFold2-predicted structures.

---

### Official Review · Reviewer_iEQ7 · 2023-11-04

**Soundness:** 2 fair
**Presentation:** 3 good
**Contribution:** 2 fair
**Rating:** 3
**Confidence:** 4

**Summary:**

Summary.

The paper is dedicated to developing approaches for protein function prediction. The authors first benchmark the retriever-based methods and predictors on protein function tasks. They further introduce EM algorithms to enable an iterative refinement between predictors and retrievers, which is called ProtIR. They show great improvements over vanilla predictor-based methods and comparable performance with protein language model-based methods.

**Strengths:**

Pros.

1. The paper is well-written and easy to follow.
2. The EM design is an interesting way to improve predictor and retriever.
3. Multiple levels of sequence similarity are considered during the empirical investigations.

**Weaknesses:**

Cons.

1. Missing important baseline and citation. "Enzyme function prediction using contrastive learning" is a recent Science paper about retrieval-based predictors for protein functions.
2. Missing the ensemble baseline. If a simple ensemble of predictor and retriever can achieve a good performance, then there is no need to do iterative refinement between predictor and retriever.
3. Missing evaluations on realistic datasets, which are referred to paper "Enzyme function prediction using contrastive learning".
4. The key advantage of retriever-based methods is the capability to annotate unseen proteins or unseen functionality. Investigations are needed to support the effectiveness of the proposed methods.
5. Beyond existing protein encoders, there are no specific biological designs for protein function annotation problems. More problem-specific characteristics like the tree structure of go terms should be considered in the modeling.
6. For the experiments, multiple runs are needed to showcase the prediction stability.

**Questions:**

Refer to the weakness section.

---

> ### Author Response · Authors · 2023-11-21
>
> Thanks for your insightful comments and great suggestions! We respond to your concerns  below:
>
> >**Q1: (1) Missing important baseline and citation. "Enzyme function prediction using contrastive learning" is a recent Science paper about retrieval-based predictors for protein functions. (2) Missing evaluations on realistic datasets, which are referred to paper "Enzyme function prediction using contrastive learning". (3) The key advantage of retriever-based methods is the capability to annotate unseen proteins or unseen functionality. Investigations are needed to support the effectiveness of the proposed methods.**
>
> Thanks for the suggestions!  We followed your suggestion to discuss more related work and include the test sets in CLEAN paper for evaluation during rebuttal. The results are shown in App. E.1, where the results of all listed baselines are included. As it is not feasible for us to fine-tune predictor-based methods on the Swiss-Prot training set within such a short time, we only consider several retriever-based methods. It can be observed that even without any supervised training on the swiss-prot training set, our proposed retrievers can outperform CLEAN on both the NEW-392 and Price-149 datasets. Also, to utilize our structure retriever on retrieving Swiss-Prot datasets, we retrieve the predicted structures from AlphaFold Database according to the UniProt ID. Although there are a small portion of proteins without structures, our structure retriever can still outperform the CLEAN method on NEW-392 dataset, which clearly demonstrates the potential of retriever-based methods.
>
> >**Q2: Missing the ensemble baseline. If a simple ensemble of predictor and retriever can achieve a good performance, then there is no need to do iterative refinement between predictor and retriever.**
>
> Thanks for the suggestion! We’ve added the ensemble baseline in App. E.3.1. As shown in Table 6, ProtIR clearly outperforms a simple ensemble baseline on all considered tasks. Only with ProtIR, can these predictors achieve comparable performance with the state-of-the-art protein language model-based methods.
>
> >**Q3: Beyond existing protein encoders, there are no specific biological designs for protein function annotation problems. More problem-specific characteristics like the tree structure of go terms should be considered in the modeling.**
>
> Thanks for the suggestion. We agree that including biological design would be helpful for protein function annotation problems. We want to argue that this will not contradict with the general idea introduced in the paper. Our strategy to train neural structure retrievers and iterative refinement framework are flexible for arbitrary protein encoders, which can include any biological design, such as GearNet and CDConv.
> As we want to focus on more general function annotation problems and make it able to generalize to unseen functions, we do not focus on function-specific characteristics like the tree structure for GO terms. We leave this direction as future work.
>
> >**Q4: For the experiments, multiple runs are needed to showcase the prediction stability.**
>
> The experiments are run on quite large training and test sets, where over 1,000 proteins are used for evaluation. For retriever-based methods, as the retrieved proteins remain the same under different random seeds, there is no need to run multiple times. For predictor-based methods, we run GearNet and CDConv on EC multiple times, where the standard errors for all these methods are below 0.003. This shows the stability of our evaluation.

---

### Official Review · Reviewer_ZTPG · 2023-11-06

**Soundness:** 2 fair
**Presentation:** 2 fair
**Contribution:** 2 fair
**Rating:** 5
**Confidence:** 3

**Summary:**

This paper proposes a new framework that can perform iterative refinement to improve the performance of predictor-based algorithm for annotating proteins' function. The main concept is to utilize EM algorithm to train both the predictor and the retriever. The authors have conducted experiments on four representative benchmarks and show that their framework is effective.

**Strengths:**

- The framework is equipped with mathematical foundation and its effectiveness can be theoretically understood.
- Understanding the function of unseen proteins is a really important task in protein engineering and other biological aplications.

**Weaknesses:**

One significant drawback of this paper is that it doesn't compare its approach to recent standard methods. This makes it difficult to understand where this new method fits in the existing research. Although the paper is in the field of functional annotation, which has a lot of existing research, the authors did not discuss relevant works. Here is a list of some papers that focus on predicting EC numbers, just to give the authors an idea:

Enzyme function prediction using contrastive learning

Deep networks for protein functional inference

ECPred: a tool for the prediction of the enzymatic functions of protein sequences based on the EC nomenclature

Deep learning enables high-quality and high-throughput prediction of enzyme commission numbers

It is important for the authors to do a fair comparison with these existing methods so that people can understand how valuable their new technique is.

------

The title of Section 5.2 should be revised to accurately reflect its content, as it encompasses benchmarking of predictor-based methods alongside other topics.

------

The impression conveyed by Table 1 is that predictor-based methods are unquestionably the superior choice for functional prediction. I fail to see evidence to support the claim "retriever-based methods show promise, enabling accurate function prediction without massive pre-training". In all evaluated tasks, predictor-based approaches with pre-training consistently outperform retriever-based methods by substantial margins.

------

The subsection titled 'Predictors vs. Retrievers' in Section 5.2 lacks clarity and fairness in its comparison. The comparison is skewed because retriever-based models like GearNet and CDConv leverage structural information that is not utilized by their predictor-based counterparts.

------

It is a time-consuming process for me to correlate Table 2 with Table 1 in order to assess the effectiveness of the proposed technique. It appears that the ProtIR framework may enhance predictor-based performance; however, the authors have not opted to demonstrate this in comparison to the best predictor-based method. Is there a particular rationale behind this choice?

------

Although Section 5.4 is intriguing, it appears to have a somewhat weak connection to the primary focus of this research.

**Questions:**

See the above section.

---

> ### Author Response · Authors · 2023-11-21
>
> Thanks for your appreciation of our work! We respond to your questions and concerns below:
>
> >**Q1: One significant drawback of this paper is that it doesn't compare its approach to recent standard methods.**
>
> Thanks for the suggestions!  We followed your suggestion to include the test sets in CLEAN paper for evaluation during rebuttal. The results are shown in App. E.1, where the results of all listed baselines are included. As it is not feasible for us to fine-tune predictor-based methods on the Swiss-Prot training set within such a short time, we only consider several retriever-based methods. It can be observed that even without any supervised training on the swiss-prot training set, our proposed retrievers can outperform CLEAN on both the NEW-392 and Price-149 datasets. Also, to utilize our structure retriever on retrieving Swiss-Prot datasets, we retrieve the predicted structures from AlphaFold Database according to the UniProt ID. Although there are a small portion of proteins without structures, our structure retriever can still outperform the CLEAN method on NEW-392 dataset, which clearly demonstrates the potential of retriever-based methods.
>
>
> >**Q2: The title of Section 5.2 should be revised to accurately reflect its content.**
>
> Thanks for the suggestions! We’ve revised the experiment section to better reflect the content and re-analyzed the experimental results. We believe the questions have been solved in Sec. 5.2.
>
> >**Q3: The impression conveyed by Table 1 is that predictor-based methods are unquestionably the superior choice for functional prediction. **
>
> Sorry for the confusion! In the revised version, we emphasize that retriever-based methods are comparable or better than predictor-based methods without pre-training. When using protein language models, predictor-based methods still outperform retriever-based methods. This can be understood because deep learning techniques efficiently leverage large pre-training datasets, enabling neural predictors to capture more evolutionary information.
>
> >**Q4: The subsection titled 'Predictors vs. Retrievers' in Section 5.2 lacks clarity and fairness in its comparison. The comparison is skewed because retriever-based models like GearNet and CDConv leverage structural information that is not utilized by their predictor-based counterparts.**
>
> Sorry for the confusion. We’ve revised our expression in the latest version.
>
> >**Q5: It appears that the ProtIR framework may enhance predictor-based performance; however, the authors have not opted to demonstrate this in comparison to the best predictor-based method.**
>
> This is a good point! One advantage of the ProtIR framework is that it can significantly enhance those predictors without massive pre-training. For PLM-based predictors, it would be expensive to do multiple refinement iterations to fine-tune these methods. Instead, we propose another method to improve PLM-based predictors by ensembling with a PLM-based retriever with structural insights, as shown in Sec. 5.4. We’ve made this point more clear in the revised version.
>
> >**Q6: Although Section 5.4 is intriguing, it appears to have a somewhat weak connection to the primary focus of this research.**
>
> Thanks for the suggestions! We rewrite the experiment section to make its logic clear. The results in Sec. 5.4 are very important and interesting, as it presents a way that we can greatly improve PLM-based retrievers by simply pre-training it with structural information.

---

> > ### Comment · Reviewer_ZTPG · 2023-11-22
> > **Response**
> >
> > I'm impressed by the authors, especially when there is such a large number of reviewers. Some of my concerns have been addressed and I decide the raise my score to 5.

---

### Comment · Area_Chair_ZSz9 · 2023-11-22
**Reviewers - Pls provide your response to authors' rebuttal**

Dear All,

The authors have dedicated significant efforts to provide detailed rebuttal. I would appreciate it if you could share your feedback with them.

Thank you for your valuable contributions to ICLR!

Best regards,
Area Chair

---

### Meta-Review · Area_Chair_ZSz9 · 2023-12-07

**Metareview:**

Scientific Claims and Findings:
The paper introduces ProtIR, a framework for iterative refinement to enhance protein function prediction. It combines retriever-based and predictor-based methods using an EM algorithm, showing significant improvements in accuracy and efficiency compared to existing techniques. The study evaluates various models on protein function tasks and demonstrates the effectiveness of ProtIR through comprehensive experiments.

Strengths:
1. Theoretical Foundation: ProtIR is equipped with a solid mathematical foundation, offering a clear theoretical understanding of its effectiveness.
2. Significance in Protein Engineering: Understanding protein function is crucial in various biological applications, and the paper addresses this important task.
3. Improvement in Performance:The framework showcases promising improvements over predictor-based methods, especially in refining predictions without massive pre-training.

Weaknesses and Missing Elements:
1. Lack of Comparative Analysis: The paper lacks direct comparisons with recent standard methods in the field, hindering a clear understanding of its position in existing research.
2. Clarity in Method Comparison: Reviewers point out a lack of clarity and fairness in comparing retriever-based and predictor-based models, especially in considering structural information.
3. Incomplete Experiment Details: Insufficient details on datasets, evaluation procedures, and specific methodological choices hinder a complete understanding of the proposed approach.

The paper shows promise in improving protein function prediction but needs to address methodological clarity, comparative analysis with existing methods, and detailed experiment descriptions for a more comprehensive and impactful contribution.

**Justification For Why Not Higher Score:**

The reviewers found the paper's main strength in its thoroughness regarding empirical results but noted several areas for improvement. One common point raised by multiple reviewers was the lack of clear comparison to existing methods in the field, which makes it difficult to gauge the novelty and significance of the proposed framework. Additionally, there were concerns about the lack of detailed descriptions of datasets, evaluation procedures, and technical aspects of the methodology, which impacted the paper's clarity.

To enhance the paper, consider addressing these specific points highlighted by the reviewers. Providing a clearer comparison with existing methods, offering more detailed explanations about datasets and methodology, and discussing the theoretical implications and potential generalizability of the proposed framework could significantly strengthen the paper.

**Justification For Why Not Lower Score:**

N/A

---

### Decision · Program_Chairs · 2024-01-16

Reject